# Observing growth and interfacial dynamics of nanocrystalline ice in thin amorphous ice films

Minyoung Lee[1,2,10], Sang Yup Lee[3,4,5,10], Min-Ho Kang [6,7,10], Tae Kyung Won[3,5], Sungsu Kang[1,2], Joodeok Kim[1,2], Jungwon Park [1,2,8,9] ✉ & Dong June Ahn [3,4,5] ✉

Ice crystals at low temperatures exhibit structural polymorphs including hexagonal ice, cubic ice, or a hetero-crystalline mixture of the two phases. Despite the significant implications of structure-dependent roles of ice, mechanisms behind the growths of each polymorph have been difficult to access quantitatively. Using in-situ cryo-electron microscopy and computational ice-dynamics simulations, we directly observe crystalline ice growth in an amorphous ice film of nanoscale thickness, which exhibits three-dimensional ice nucleation and subsequent two-dimensional ice growth. We reveal that nanoscale ice crystals exhibit polymorph-dependent growth kinetics, while hetero-crystalline ice exhibits anisotropic growth, with accelerated growth occurring at the prismatic planes. Fast-growing facets are associated with low-density interfaces that possess higher surface energy, driving tetrahedral ordering of interfacial $H_2O$ molecules and accelerating ice growth. These findings, based on nanoscale observations, improve our understanding on early stages of ice formation and mechanistic roles of the ice interface.

Ice crystallization is a ubiquitous process having significant implications in various scientific and technological fields[1-7]. The formation of structural polymorphs, including amorphous and crystalline forms of ice are influenced by conditions such as temperature, pressure, and preparation protocols[4,8-19]. Hexagonal ice (ice $I_h$), the most common crystalline polymorph, is widely found in atmospheric conditions[11,15]. Cubic ice (ice $I_c$) is another crystalline polymorph reported to coexist with ice $I_h$ at a broad temperature range of 160–240 K[13,15]. Despite the marginal difference in the thermodynamic favorability for the growths of ice $I_c$ and $I_h$[20,21], these polymorphs exhibit distinct kinetic growth properties[22,23]. This implies that there are other important factors that influence the nucleation and growth of ice crystals. Considering the important role of interfaces in the nucleation and growth of solids[24-29], factors affecting ice growth are also likely relevant to interfacial characteristics of ice polymorphs[12,13]. Indeed, there had been observations in which ice growth shows a dependency on the nature of interfaces in the environment. For instance, conditions within nanopores[30] or nanoscale water droplets[22,31] have favored the formation of ice $I_c$ structures over $I_h$.

[1]School of Chemical and Biological Engineering, and Institute of Chemical Processes, Seoul National University, Seoul 08826, Republic of Korea. [2]Center for Nanoparticle Research, Institute of Basic Science (IBS), Seoul 08826, Republic of Korea. [3]Department of Chemical and Biological Engineering, Korea University, Seoul 02841, Republic of Korea. [4]KU-KIST Graduate school of Converging Science and Technology, Korea University, Seoul 02841, Republic of Korea. [5]The w:i Interface Augmentation Center, Korea University, Seoul 02841, Republic of Korea. [6]Department of Biomedical-Chemical Engineering, The Catholic University of Korea, Bucheon-si 14662, Republic of Korea. [7]Department of Biotechnology, The Catholic University of Korea, Bucheon-si 14662, Republic of Korea. [8]Institute of Engineering Research, College of Engineering, Seoul National University, Seoul 08826, Republic of Korea. [9]Advanced Institutes of Convergence Technology, Seoul National University, Suwon-si 16229, Republic of Korea. [10]These authors contributed equally: Minyoung Lee, Sang Yup Lee, Min-Ho Kang. ✉e-mail: jungwonpark@snu.ac.kr; ahn@korea.ac.kr

Although numerous studies, mainly based on diffraction techniques, have attempted to investigate the growth mechanisms of ice at low temperatures[13,15,18,19,32–39], it has been extremely challenging to understand polymorph-dependent growth dynamics of ice crystals and the role of interfacial structures for the observed crystallization processes. This is because of the difficulty in producing pure ice polymorphs for independent investigation. Thus, many of previous studies are limited in elucidating how coexisting ice $I_c$ and $I_h$ sequences are grown and manifested in crystallized domains[34,40,41]. Experiments that aimed to produce ice $I_c$ for diffraction analysis, including freezing water[37], warming amorphous ice[32,36,42], or dissociating gas hydrates[43] have instead resulted in what is now referred to as stacking-disordered ice, which is a heterogeneous mixture of cubic sequences interlaced with hexagonal sequences[23,37,38,44,45]. Only recently has pure cubic ice been produced experimentally by heating ice XVII[18] and degassing hydrogen from hydrogen hydrate[19] at scales observable with diffraction techniques. Another challenge lies in that diffraction techniques provide spatially averaged information, so behaviors of different ice polymorphs and contributions from different crystal facets are not deconvoluted. Separately probing individual ice polymorph physics, including local crystal structures and interfaces, provides insight into polymorph-dependent growth mechanisms of ice and the roles of heterocrystals or defects. In this report, we use in-situ cryo-electron microscopy (cryo-EM) and molecular dynamics (MD) simulations for modeling growth at different ice surfaces to directly observe the time-resolved growth of individual ice nanocrystal polymorphs on amorphous ice films with nanoscale thickness and investigate their respective interfacial structural dynamics. Our results reveal that nanoscale ice crystals in the early stage of growth exhibit polymorph-dependency, with hetero-crystalline ice exhibiting anisotropic growth and expedited growth in the direction of prismatic planes. We further elucidate that fast-growing facets are associated with lower density of water molecules at interfaces that possess higher surface energy, driving tetrahedral ordering of interfacial $H_2O$ molecules and hence accelerating ice growth.

## Results and discussion
### Early-stage crystallization of ice nanocrystals
Nanoscale growth processes of ice crystals within suspended amorphous ice films were investigated using temperature-controlled cryo-EM. Amorphous ice was prepared by plunge-freezing a thin aqueous film on a TEM grid, which was transferred to a cryo-transfer holder (Fig. 1a and Methods). The average thickness of the free-standing amorphous ice, determined by energy filtered TEM, is 132 nm with a standard deviation of 13.4 nm (Supplementary Fig. 1), whereby crystallization kinetics is not thickness-dependent[46–49]. Crystallization of the amorphous ice was induced by ramping the cryo-transfer holder temperature, and the growths of ice crystals were investigated at 143 K where crystallinity started to emerge in selected area electron diffraction (SAED) patterns (Supplementary Fig. 2). A low-dose imaging method was used, in which images of consecutive holes were acquired sequentially while the holder temperature was maintained at 143 K, yielding images of ice films at different times during annealing (Fig. 1b and Methods). With this process, we observed heating-induced crystallization of amorphous ice, free from beam-induced crystallization of amorphous ice films that usually occurs in the prolonged exposure of ice films to irradiation (Supplementary Figs. 3, 4 and Supplementary text 1). As shown in the bright-field TEM (BFTEM) images of ice undergoing heating-induced crystallization, individual ice domains with dark contrast appear as soon as the holder reaches 143 K. The dark contrast originates from the transformation of amorphous ice into crystalline domains. This was revealed through identical-location TEM, in which a hole was imaged once at 93 K and again after the temperature was ramped to 143 K (Supplementary Fig. 5a). The images of amorphous ice before the formation of crystalline domains show

uniform contrast without high-contrast features that may be indicative of any contaminants. The holes with adsorbed ice contaminants were excluded from quantification (Supplementary Fig. 5b). Additionally, through spectroscopic elemental analysis, we rule out the possibility of the dark contrast domains being contaminants (Supplementary Fig. 5c–f and Supplementary text 2). Accordingly, the dark contrast features originate from the crystallinity of the ice domains formed and they are clearly delineated in the bright-field TEM where the contrast of the ice domains is further enhanced (Supplementary Fig. 6).

The dark contrast ice domains increase in number and change in area over time and consume the amorphous ice completely within 907 s (Fig. 1c). Growth kinetics of ice nanocrystals were investigated by measuring their areas through a contrast enhancement procedure (Supplementary Fig. 7 and Methods) and evaluating the number of domains and the crystallized fraction of amorphous ice over time (Methods). The number of ice domains increases for the first 215 s, and then decreases slightly at later annealing times (Fig. 1d). The crystallized fractions over time in Fig. 1e exhibits a characteristic sigmoidal profile described by the Avrami equation:[50]

$$X_c = 1 - e^{-\left(\frac{t}{\tau}\right)^n} \qquad (1)$$

fitted with $\tau = 354.6$ s and $n = 2.88$. While the amorphous ice film is estimated to be at a scale thick enough to consider the nucleation and crystallization as a bulk process, we note that once an ice domain's diameter exceeds around 132 nm, the thickness of the ice film, ice growth along the thickness axis of the ice film stops, but continues growth along the directions normal to the thickness (Supplementary Fig. 8a). This starts to occur after approximately 215 s of annealing, where ice nanocrystal diameters start to increase rapidly beyond around 300 nm (Supplementary Fig. 8b), becoming the dominant mechanism of crystallization. Initially, the fitted Avrami parameter is $n = 0.613$ indicating 3D-ice nucleation of the ice particles, and after 215 s the fitted Avrami parameter becomes $n = 2.98$, based on the log-log plot in Supplementary Fig. 8c, which indicates 2D-ice growth[50]. Based on these results, we can conclude that, during crystallization, nanocrystals first nucleate within amorphous ice, followed by their growth until around 132 nm, and if their diameters reach beyond 132 nm they would grow large enough to impinge on the surface and their growths along the thickness axis will be halted. The sigmoidal profile of the overall Avrami equation models three stages of growth; the first stage is characterized by the slow increase in the crystallized fraction, the second stage by an acceleration in crystallization rate due to rapid growth of domain sizes, and the third stage by a plateau phase, where ice growth approaches saturation and is slowed down. As mentioned previously, the first stage corresponds to three-dimensional ice nucleation, while the second and third stages correspond to two-dimensional ice growth. Histograms of particle areas at each of the nucleation, rapid-growth, and plateau stages depict the statistics of individual domain areas that comprise the crystallized films at each stage (Fig. 1f). Larger domains increase in count at the rapid-growth and plateau stages. Surprisingly, counts of small domains with areas below 7000 nm² also persist in completely crystallized ice films, suggesting the presence of a type of domain that remains small with relatively slowed growth. The histograms of the rapid-growth and plateau stages were fit with two Gaussian functions, revealing that the distribution for the smaller domains were centered at 3500 nm² and included particles up to 7000 nm², while the larger domains had broader distributions of domain areas centered at 9000 nm² and 9500 nm² for the rapid-growth and plateau phases, respectively. These results verify that one type of domain remains a small size due to slow growth and does not continuously grow into large domains, while another type of domain keeps growing to a larger size until the amorphous ice is fully crystallized.

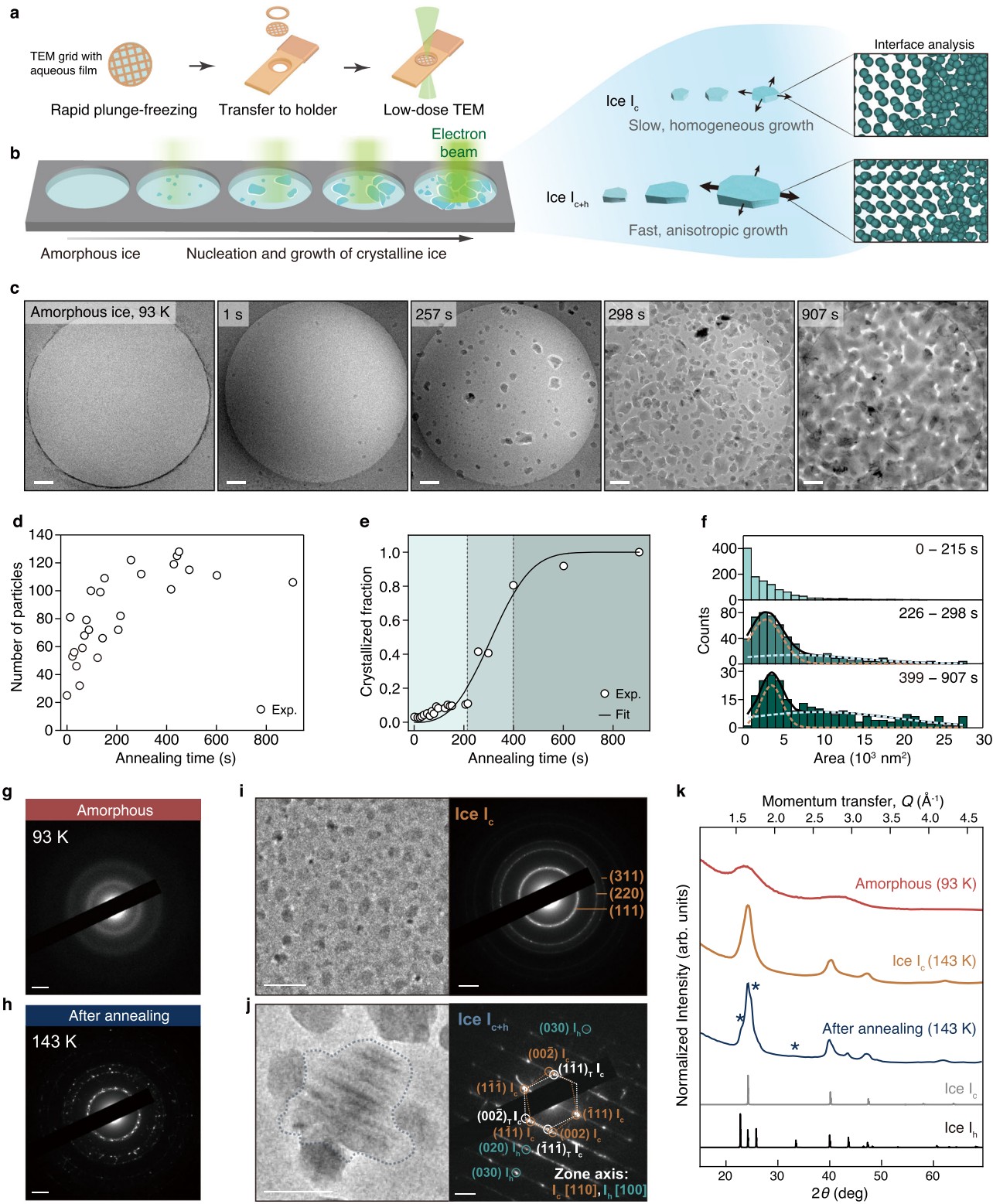

Selected-area electron diffraction (SAED) was used to verify the crystal phases of the domains. SAED patterns obtained from amorphous ice at 93 K and ice after annealing at 143 K, as shown in Fig. 1g, h, respectively, demonstrate that annealed ice results in the appearance of multiple crystalline domains. To analyze the phases of the small, slow-growing ice nanocrystals, SAED was acquired at a position consisting solely of small ice domains smaller than 7000 nm² (Fig. 1i and Supplementary Fig. 9). The SAED patterns clearly reveal ring structures at the positions of (111), (220) and (311) peaks of cubic ice (ice $I_c$)[51,52].

Additionally, the SAED patterns of individual small ice nanoparticles reveal single-crystal peaks of ice $I_c$ (Supplementary Fig. 10a). Larger ice nanocrystals, as shown in Fig. 1j and Supplementary Fig. 10b, contain nanoscale defects manifested by streaks in both SAED and nanobeam diffraction (NBD) patterns, which also exhibits peaks for twin ice $I_c$ planes as well as ice $I_h$ sequences. These streaks, which are not present for pure ice $I_c$ or $I_h$ sequences, are consistent with the streaks shown in electron diffraction simulations of hetero-nanocrystalline ice (Supplementary Fig. 11 and Methods). Radial averages of the SAED patterns

**Fig. 1 | Growth of nanocrystalline ice from amorphous ice. a** A schematic of the method for producing amorphous ice films, in which a nanoscale aqueous ice film is plunge-frozen, transferred to a cryo-transfer holder held at 93 K, and placed in a TEM where the temperature was ramped to 143 K. **b** A schematic for characterizing the growth and interfacial structures of nanocrystalline ice from amorphous ice films with low electron dose. During annealing of the ice film at 143 K, each hole is imaged once before the beam is moved on to the next hole. The experiment allowed the characterization of phases of ice and the associated growth properties. **c** Representative images of amorphous ice at 93 K, and ice undergoing crystallization at 143 K with increasing annealing time. **d** Number of ice particles formed from a total of 26 images, with each data point obtained from each image. **e** Crystallized fraction over time, obtained from a total of 22 images. The Avrami equation is fitted with $\tau$ = 354.6 s and $n$ = 2.88. The shading indicates the stages of crystallization including nucleation (light green), growth (moderate green), and the plateau phase (dark green). **f** Histogram of ice particle sizes over different time frames, with each histogram corresponding to regions indicated by different color shades in **e**. The dotted lines represent Gaussian fits, with the orange lines centered at 3500 and the light blue lines centered at 9000 and 9500 for the middle and bottom histograms, respectively. The solid black line represents the sum of the two Gaussian fits. **g** SAED pattern of amorphous ice at 93 K and **h** SAED pattern of ice annealed at 143 K, from the same sample but different region as in **g**. **i** TEM image (left) and SAED pattern (right) of small, finitely growing ice $I_c$ nanocrystals and **j** TEM image (left) and nanobeam diffraction pattern (right) of a large, continuously growing ice $I_{c+h}$ nanocrystal exhibiting hetero-crystallinity and planar defects, shown by the presence of $I_h$ features and streaks in the diffraction pattern. The dotted line in the TEM image delineates an ice nanocrystal domain of interest. **k** Radially averaged SAED patterns for amorphous ice at 93 K in **g** (red), ice $I_c$ in **i** (orange), and ice after annealing at 143 K in **h** (blue). The asterisks mark the unique peaks of ice $I_h$. Simulated XRD data for ice $I_c$ (gray) and $I_h$ (black) are plotted with $2\theta$, which were calibrated from the momentum transfer values. Scale bars for TEM images = 100 nm. Scale bars for diffraction patterns = 2 nm⁻¹.

of Fig. 1g–i were compared with XRD simulations of $I_c$ and $I_h$ structures to gain further understanding of ice nanocrystal phases grown in amorphous ice (Fig. 1k). Small ice domains exhibiting finite growth are purely ice $I_c$, as further confirmed by the absence of the (112) as shown in the XRD simulations. In contrast, annealed ice containing large ice nanocrystals consists of ice $I_h$ peaks, namely the shoulder peaks before and after the most intense peak at 1.65 Å⁻¹ and the (112) peak at 2.95 Å⁻¹. The relatively high intensity of the peak at 1.65 Å⁻¹ compared to its shoulder peaks indicates that fast-growing larger ice domains consist of both $I_c$ and $I_h$ sequences and are hetero-nanocrystalline (namely, ice $I_{c+h}$).

## Molecular structure of the hetero-crystalline ice domain

Molecular-scale TEM imaging of the ice $I_{c+h}$ domain at the [110] zone axis was performed to investigate the hetero-crystalline structure and to reveal the structural basis on the fast-growing propensity. A high magnification TEM image of an ice $I_{c+h}$ domain is shown in Fig. 2a. Extracted from the high-magnification image of ice $I_{c+h}$, the FFT of a cubic stacking region marked with a box with dashed orange lines in Fig. 2a contains peaks that correspond to the ice $I_c$ structure at the [110] zone axis (Fig. 2b) while the FFT from the box with dashed blue lines in Fig. 2a exhibits vertical streaks that are characteristic of stacking disorder (Fig. 2c). The inverse FFT (iFFT) prepared by masking the ice $I_c$ ($\bar{2}20$) peaks from the boxed region with dashed white lines in Fig. 2a indicates the regions containing cubic sequences within the ice $I_{c+h}$ hetero-nanocrystal, as shown in Fig. 2d. Dark-colored regions in the iFFT do not correspond to $I_c$ sequences and may indicate either ice $I_h$ sequences or stacking defects. Homogeneous sequences of ice $I_c$ and $I_h$ incorporated in the crystal normally exhibit uniform contrast in the TEM image, while stacking faults involve a change in contrast compared to the surrounding layers. Close-up views of molecular arrangements at the regions boxed with orange and blue lines in Fig. 2a are shown in Fig. 2e, f, respectively. Models of ice $I_c$, ice $I_h$, and ice $I_{c+h}$ were generated using MD simulations (Fig. 2g–i). The models and the TEM simulations show similar images to the local structures visualized with the high-magnification TEM images (Fig. 2j–l and Methods). Additionally, extensive TEM simulations that match our experimental imaging conditions were performed for different thickness and defocus values of the proposed structures of ice at a region with a stacking fault, as discussed in detail in Supplementary text 1.3 and Supplementary Fig. 12. The TEM simulation results verify that the atomic positions are manifested by light contrast in our imaging conditions. With this structure, the {111} face of $I_c$ and the basal face of the $I_h$ structure are equivalent, allowing for the interlacing of $I_c$ and $I_h$ sequences stacked in the direction normal to the basal plane. In addition, the primary and secondary prismatic planes of ice $I_h$ sequences correspond to {110} and {112} planes of $I_c$, respectively (Fig. 2m, n). Along the direction normal to the basal plane, the hetero-

crystalline structure contains a combination of $I_c$ and $I_h$ stacking sequences (Fig. 2o). This structure of ice consisting of both ice $I_c$ and ice $I_h$ sequences with stacking disorder was observed for multiple fast-growing ice domains in our HRTEM experiment as shown in Supplementary Fig. 13, which has been proposed experimentally[12,37] and by computational modeling[23,38,44,45] in previous studies.

## Growth dynamics of ice nanocrystal polymorphs

The growths of individual ice $I_c$ and ice $I_{c+h}$ nanocrystal domains were tracked with in-situ imaging by obtaining time-series images of ice particles growing on the amorphous ice film (Supplementary Movie 1 and Methods). TEM images were acquired with the electron beam blanked between consecutive frames, allowing for minimal exposure of the ice film to the electron beam. Ice $I_c$ particles labeled from P1 to P5 in Fig. 3a exhibit relatively slow growth over the observation period, as shown by the plot of the change in area over time in Fig. 3b. Meanwhile, ice $I_{c+h}$ domains exhibit continuous growth on the amorphous ice film. Some of these domains exhibit dark streaks which indicate the direction of planar stacking faults and thus allow us to define the growth directions of basal planes and prismatic planes (Fig. 3c and Supplementary Fig. 14). Interestingly, the prismatic planes exhibited faster growth than the basal planes, revealing the anisotropy among the growth rates of the two planes. The prismatic plane of the particle in Fig. 3c was observed to have a higher initial growth rate than the basal plane by 1.56 times, with the displacement of the prismatic plane eventually doubling the displacement of the basal plane (Fig. 3d).

Tracking the areas of 64 crystalline domains over time, we verified the dynamics of the slow and fast growth propensities of ice $I_c$ and ice $I_{c+h}$, respectively. The slow and fast growths of ice $I_c$ and ice $I_{c+h}$ domains, respectively, are apparent by the difference in the extents of growth of each type of domain, measured from the time-resolved TEM observations in Fig. 3e. The domains of ice $I_c$ (shown in orange) are initially small, exhibiting relatively little growth until the final frame. By contrast, the domains of ice $I_{c+h}$ (shown in blue) are larger and expand mostly at steeper rates than those characterized as $I_c$. The distributions of growth rates against time for ice $I_c$ and $I_{c+h}$ nanocrystals are quantified in Fig. 3f and g, respectively, revealing their distinct growth dynamics. Growth rates of ice $I_c$ nanocrystals are smaller and relatively more constant than those of ice $I_{c+h}$. This implies that the growth dynamics of ice $I_c$ are less affected by the exhaustion of the amorphous phase. On the other hand, ice $I_{c+h}$ crystals exhibit a variety of areal growth rates which decrease as time passes. This is attributed to the increase in crystallized fraction of the amorphous ice film resulting in the impingement of ice $I_{c+h}$ domain interfaces with other adjacent domains, slowing down their growth (Supplementary Fig. 15). A statistical analysis on the aspect ratios of ice $I_c$ and $I_{c+h}$ is shown in the histograms in Fig. 3h, i, respectively. A larger fraction of ice $I_{c+h}$ nanoparticles has higher aspect ratios compared to ice $I_c$. This further

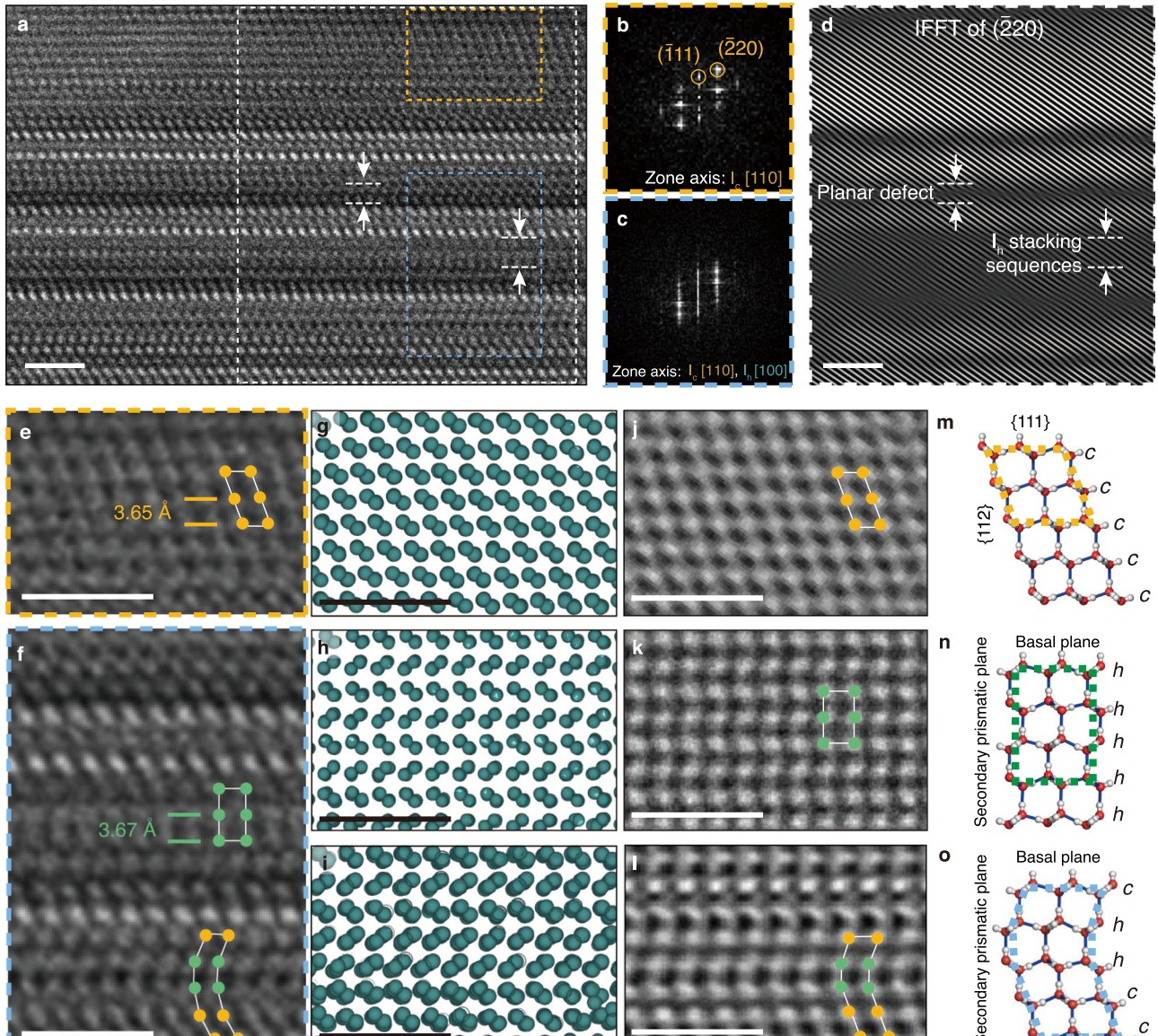

**Fig. 2 | The structure of the ice $I_{c+h}$ domain. a** HRTEM image of a grown hetero-crystalline ice $I_{c+h}$ domain at the [110] zone axis. FFT of the boxed areas in **a** at a region of ice $I_c$ sequences along [110] zone axis (**b**), and a hetero-crystalline region (**c**). **d** IFFT of the ($\bar{2}$20) peak of the white boxed region in **a**, revealing positions of non-cubic sequences. **e**, **f** Zoomed-in, low-pass filtered images of the region marked with the orange dotted line showing $I_c$ sequences (**e**), and with the blue dotted line showing $I_h$ sequences and stacking defects (**f**). Models constructed for ice $I_c$ (**g**), ice $I_h$ (**h**), and ice $I_{c+h}$ (**i**) sequences, in which the turquoise spheres represent oxygen atoms. TEM simulations results of ice $I_c$ (**j**), ice $I_h$ (**k**), and ice $I_{c+h}$ (**l**) models. Configurations of ice $I_c$ (**m**), ice $I_h$ (**n**), and ice $I_{c+h}$ (**o**) crystal structures with labeled planes. Scale bars = 2 nm.

demonstrates the anisotropic nature of the growth of ice $I_{c+h}$ along different crystallographic planes.

All atomic molecular dynamics (AAMD) simulations were performed to elucidate the molecular structures of $H_2O$ molecules at the solid-liquid interface during crystallization. We implemented an ice growth system using three-dimensional crystal seeds to calculate growth rates based on the arrangement of $H_2O$ molecules on the ice surface (Methods). The simulations were conducted at 233 K, under conditions in which ice seeds with high curvature can grow, and water molecules possess adequate mobility for phase transitions. We note that higher temperatures increase the extent of water molecule diffusion and consequently the ice growth rate, and therefore, a higher growth rate of the ice crystal is expected compared to our experimental conditions (143 K). The simulations were modeled and interpreted as an indication of how different ice surfaces undergo crystallization. Growth rates for each plane (i.e., facets corresponding

to basal and secondary prismatic planes) of ice $I_c$, ice $I_h$, and hetero-crystalline ice $I_{c+h}$ were calculated from simulations. As shown in Fig. 3j, ice $I_c$ exhibited consistent growth along different planes, which resulted in an isotropic crystal. Meanwhile, ice $I_h$ shows anisotropic growth, with the prismatic planes growing more rapidly than the basal plane. Notably, in hetero-crystalline ice, $I_{c+h}$, anisotropic ice growth was most prominent, while areal growth also exceeded those of $I_c$ or $I_h$. To quantify the extent of growth, we tracked the positions of the ice surfaces over time (Fig. 3k). Displacements of the {111} plane of ice $I_c$ and basal planes of ice $I_h$ and ice $I_{c+h}$ along their growth directions were relatively similar, with values of 1.40, 1.35 and 1.53 nm, respectively. In contrast, those of {112} of ice $I_c$ and the secondary prismatic planes of ice $I_h$ and ice $I_{c+h}$ differ to a larger extent, having values of 1.59, 2.87 and 3.83 nm, respectively.

Through direct observation with in-situ cryo-EM, we have observed that ice $I_c$ exhibits markedly slow growth rates compared

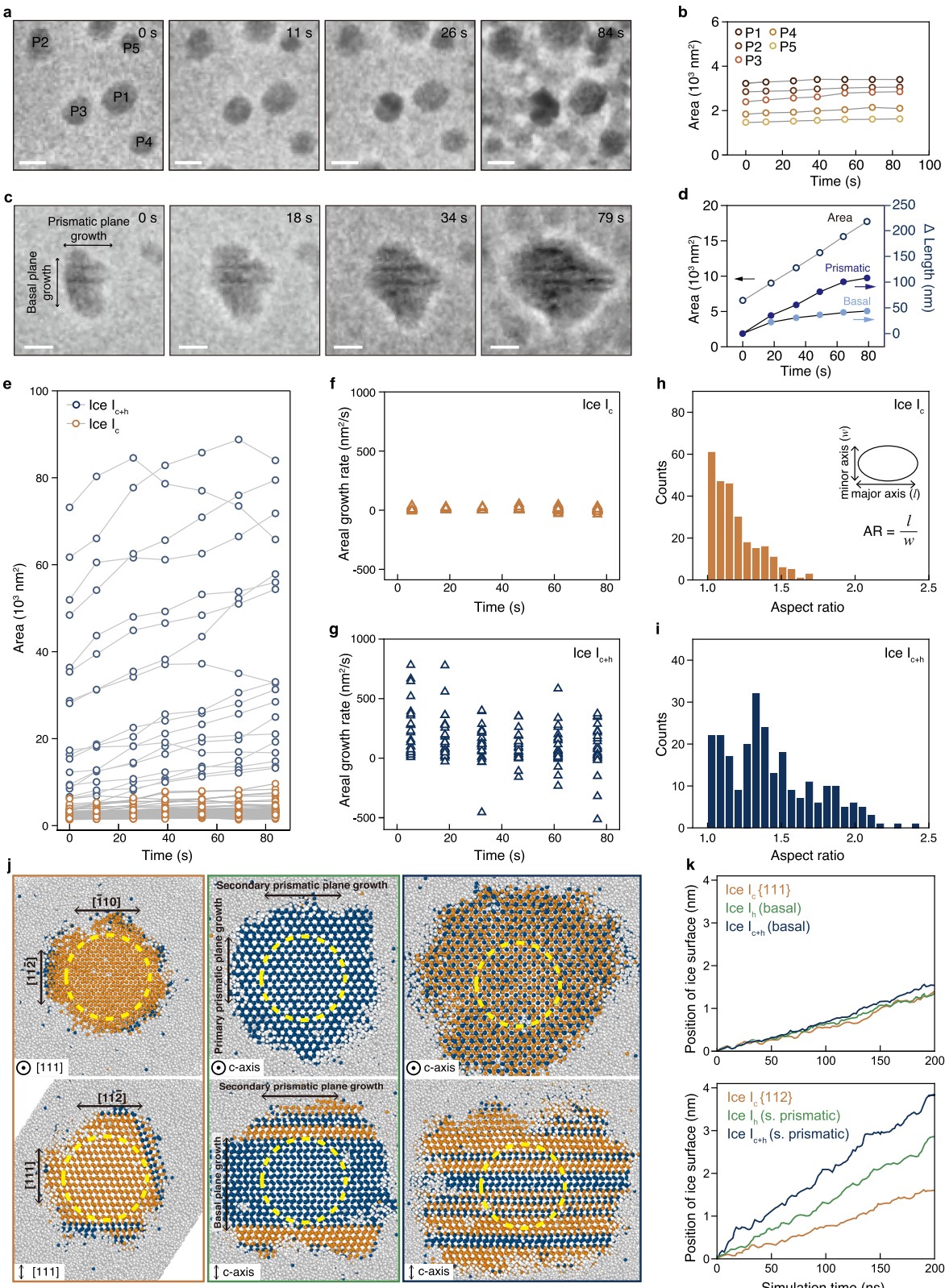

to ice $I_{c+h}$, which exhibits faster growth at the prismatic planes compared to the basal planes. MD simulations have consistently confirmed that hetero-crystalline ice $I_{c+h}$ exhibited faster crystal growth rates than homogeneous ice on the basis of enhanced prismatic plane growth. Notably, a computational study has revealed that stacking disorder results in a decrease in free energy due to entropic stabilization of larger ice crystals and hence,

nucleation rates have increased by incorporating stacking dis-order in ice[23]. These observations are consistent to what we infer from the results of our study in terms of the kinetically favored formation of crystals possessing defects, as the hetero-crystals associated with numerous defects exhibit defect-dependent growth at the interfaces which occurs prominently along the prismatic planes.

**Fig. 3 | Growth dynamics of ice nanocrystal polymorphs. a** Time-series images of five representative ice $I_c$ nanocrystals, labeled P1 through P5, exhibiting limited growth (Scale bars = 50 nm) and **b** changes in their areas. **c** A continuously-growing ice $I_{c+h}$ nanocrystal with planar defects and **d** the change in its area (empty circles) and lengths (solid circles) along the growth directions of basal and prismatic planes. **e** Changes in area plotted over time for 64 ice $I_c$ (orange) and ice $I_{c+h}$ (dark blue) nanocrystals. Growth rates over time for ice $I_c$ (**f**) and ice $I_{c+h}$ (**g**). Histogram of aspect ratios (AR) summed over time for ice $I_c$ (**h**), and ice $I_{c+h}$ (**i**). **j** Cross-section images of ice growth simulation from spherical seeds of ice $I_c$ (first column), ice $I_h$

(second column), and ice $I_{c+h}$ (third column) calculated for 200 ns at a temperature of 230 K. The first and second rows display top view images along the [111] or basal axis and side view images perpendicular to the axes, respectively. Oxygen atoms in water molecules are represented by the following: cubic structure in orange, hexagonal structure in dark blue, and other water molecules as gray spheres. The initial position of ice seed with 5 nm diameter is marked by a yellow dashed line. The arrows indicate the direction of growth. Scale bar = 2 nm. **k** Temporal tracking of ice plane positions for ice $I_c$ (yellow), ice $I_h$ (green), and ice $I_{c+h}$ (dark blue).

## Role of quasi-ice interfacial structure on growth dynamics

We further investigated the role of the interfacial regions of ice $I_c$ and $I_{c+h}$ nanocrystals and the effects on the anisotropic growth dynamics of different facets. Time-series TEM images with false-color maps of representative particles of ice $I_c$ and ice $I_{c+h}$, labeled P1 and P2 respectively, are shown in Fig. 4a. P2 exhibits bright-contrast surroundings as shown by the light-yellow areas while P1 does not have these features. Line profiles of the images for P1 and P2 reveal differences in the contrast of the surroundings of the particles (Fig. 4b). The bright-contrast surroundings were unique for ice $I_{c+h}$ and not present in ice $I_c$ domains, as shown in multiple TEM images of ice nanocrystals (Supplementary Fig. 16a–d). In addition, the HAADF-STEM image of an ice $I_{c+h}$ particle verifies a difference in contrast in the interfacial regions and reveals that the contrast originates from low mass-thickness and is attributed to the formation of a mass-depleted region in the amorphous ice (Supplementary Fig. 16e). The contours of P1 and P2 shown in Fig. 4c and the changes in their area over time plotted in Fig. 4d verify the contrasting growth rates of the respective ice $I_c$ and ice $I_{c+h}$ domains. This demonstrates that there are significant differences in the thickness, ice densities, or structures of the surroundings and the growth interface, which are associated with the slow and fast growth rates of ice $I_c$ and ice $I_{c+h}$ ice domains, respectively.

A high-resolution TEM image that includes the surroundings of a heterocrystalline ice domain is shown in Fig. 4e. The atomic columns of white contrast indicate crystallinity from the ice domain, which exhibit $I_c$ stacking sequences as well as planar defects. These atomic columns extend up to the amorphous interfacial region and decrease in contrast near the interfacial region. The inverse FFT image in Fig. 4f for the masked crystalline peaks (inset) reveal the region occupied by crystalline ice where pixel intensities are bright and color-mapped in yellow, and the interfacial region that lacks crystallinity which is closer to purple. Electron diffraction, a method that can be used to probe amorphous ice density[53] was performed to investigate the structure of the surrounding ice at the interfacial region of a fast-growing ice nanocrystal, which is otherwise difficult to interpret with TEM images alone (Supplementary Fig. 17 and Supplementary text 1.4). We obtained a SAED pattern from the interfacial region of a representative ice $I_{c+h}$ particle shown in Fig. 4g, placing the 150-nm SA aperture at a region containing a growing ice $I_{c+h}$ particle and the interfacial region surrounding it. We then compare this to the SAED of amorphous ice at 143 K that had not undergone crystallization. The amorphous halo ring from the SAED pattern of the interfacial region shows a shift inward compared to the pattern of amorphous ice at 143 K (Fig. 4h). The amorphous components excluding the crystalline peaks of the SAED patterns were radially averaged and the background was subtracted. The first peak of the SAED pattern from amorphous ice at 143 K is at 1.77 $Å^{-1}$, which is shifted to a higher reciprocal distance compared to the 93 K low-density amorphous ice (LDA) peak at 1.65 $Å^{-1}$ (Fig. 1g). This is attributed to a transition that would be manifested by the increase of observed density of the amorphous ice before undergoing crystallization, consistent to some reports that mention the transformation of amorphous ice into a viscous form above 136 K[54,55]. Meanwhile, the first peak from the ice $I_{c+h}$ growth interface exhibits a shift to lower

reciprocal distance compared to the amorphous phase at 143 K from 1.77 to 1.69 $Å^{-1}$ (Fig. 4i and Supplementary Fig. 18). Such shift is indicative of a decrease in the density of ice[56,57], suggesting that the amorphous region near a growing ice $I_{c+h}$ crystal is lower in density compared to amorphous ice at 143 K. In addition, the decrease in the full width at half maximum (FWHM) of the first peak of the interfacial region suggests an increase in molecular order and indicates that the amorphous region near the interface of a continuously-growing ice nanocrystal starts to structurally resemble the characteristics of crystalline ice.

We performed AAMD simulations to scrutinize the molecular structure of the low-density regions that arise with the growth of ice $I_{c+h}$ crystals. As shown in Fig. 5a, the density profiles of water molecules in the freezing direction were calculated for ice $I_c$, ice $I_h$, and ice $I_{c+h}$. Low-density regions with less than 1 g/ml were observed at the interface of prismatic planes of ice $I_h$ and ice $I_{c+h}$. In the coarse-grained density profile using a thick slab (5 Å), which excludes the density peaks and valleys caused by the ice lattice, these low-density regions were also observed (Supplementary Fig. 19). In contrast, low-density regions did not appear on basal planes nor on any of the planes of ice $I_c$, which were slow-growing surfaces as was established earlier. Subsequently, we analyzed the radial distribution of interfacial $H_2O$ molecules according to different crystal facets (Fig. 5b). For facets of ice $I_c$, solvation peaks at 2.8 Å and 4.5 Å were observed, which correspond to hydrogen bonds in the first and second coordination shells of liquid water. For the interfacial molecules of prismatic planes of ice $I_h$ and $I_{c+h}$, an ice-specific peak emerges at 5.3 Å. This peak position is characteristic of the tetrahedral hydrogen bonding network in crystalline ice structures. When these results were compared to radial distribution functions (RDF) obtained using empirical potential structure refinement simulations, we discovered that the RDF outcomes for interfacial water molecules of ice $I_c$ {112} planes resemble the simulated RDF of high-density liquid (HDL) while those of prismatic planes of ice $I_h$ and $I_{c+h}$ resemble the simulated RDF of low-density liquid (LDL)[58]. Moreover, we calculated the tetrahedral order parameter for ice, bulk liquid water, and the interfacial $H_2O$ (Fig. 5c). Ice exhibits a value close to 1.0 due to the highly ordered $H_2O$ molecules, whereas liquid water has a value of 0.55. The interfacial $H_2O$ of ice $I_c$ showed an order value akin to that of liquid water, indicating the presence of a disordered structure lacking tetrahedral hydrogen bonding. Interestingly, on both the primary and secondary prismatic planes of ice $I_h$ and $I_{c+h}$, the simultaneous presence of water molecules exhibiting both ice-like ordered structures and liquid water-like disordered configurations were observed. Such ordered structures at the interfacial region result in decreased $H_2O$ mobility[59,60] which promotes the attachment of molecules onto the nanocrystal and accelerates the growth of the corresponding facet. We note that the interfacial energies at the {112} plane of ice $I_c$, and the prismatic planes of $I_h$, and $I_{c+h}$ were determined to be 22.1 $mJ/m^2$, 23.6 $mJ/m^2$ and 25.4 $mJ/m^2$, respectively (Supplementary Fig. 20 and Supplementary Table 1). Through the investigation of water molecule structures at the interface including the analysis of radial distributions and tetrahedral order parameters and through determining the interfacial energies, we found that the low-density ordered

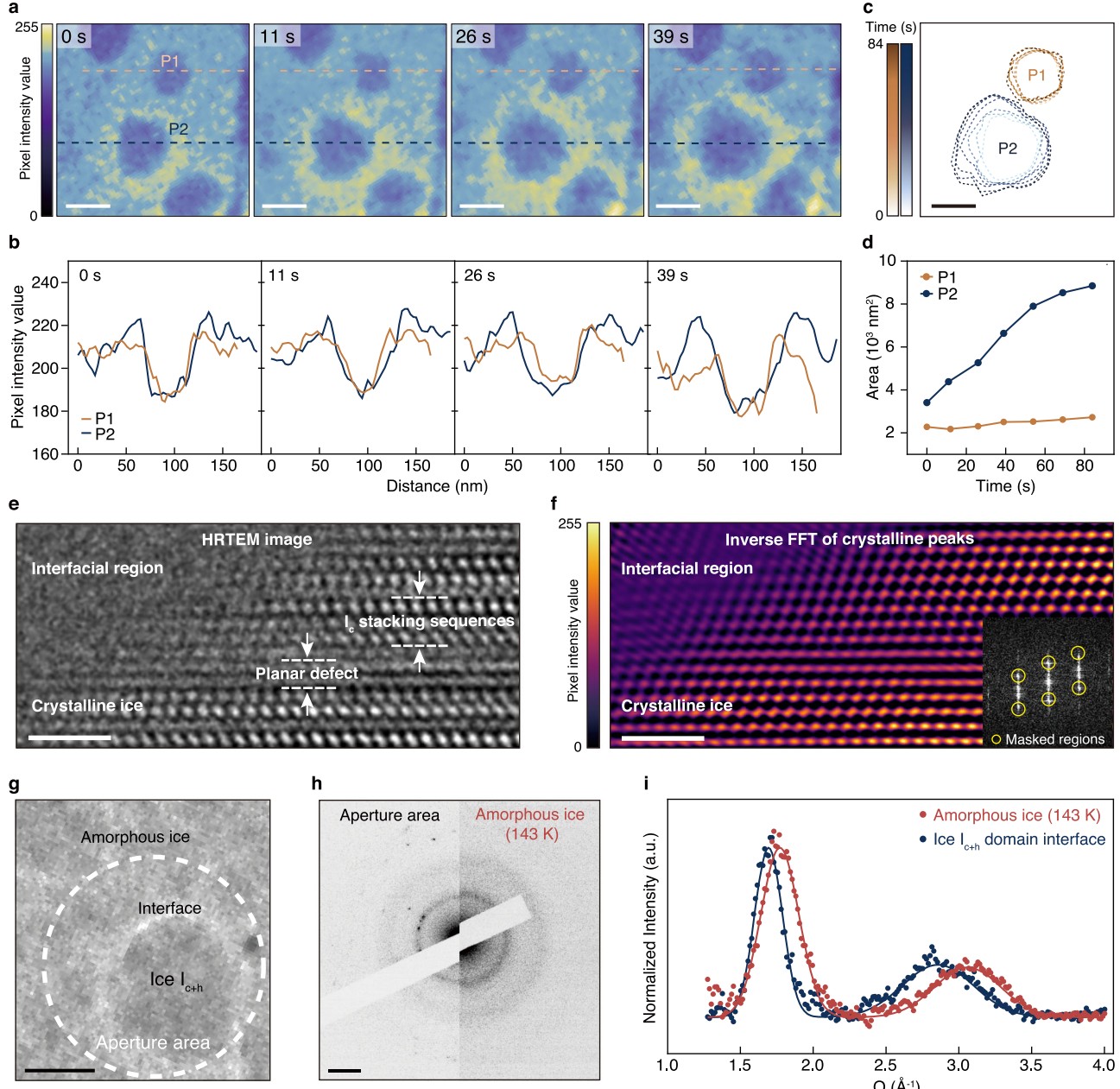

**Fig. 4 | The crystalline/amorphous ice interface. a** False color time-resolved in-situ TEM images of the first four frames of the growths of P1 and P2, which correspond to ice $I_c$ and ice $I_{c+h}$ nanocrystals, respectively. Scale bar = 50 nm. **b** Intensity profiles of P1 and P2 in sequential frames of the TEM images. P2 shows the presence of a bright-contrast region indicative of low mass-thickness contrast. **c** Contours of P1 and P2 and **d** changes in their area over time. Scale bar = 50 nm. **e** HRTEM image of a heterocrystalline ice domain that includes the surroundings of the domain. Stacking sequences are labeled. **f** False-colored inverse FFT image, which shows areas that are high in crystallinity as yellow, and areas that are low in crystallinity in purple. Scale bars = 2 nm. **g** TEM image of a representative continuously-growing ice domain with its interfacial region. The white dashed circle represents the area where the aperture was inserted for SAED analysis. Scale bar = 50 nm. **h** Diffraction patterns obtained at the aperture area and at amorphous ice at 143 K. **i** Radial average of the amorphous components of the diffraction patterns in **h**, with the data points (solid dots) fitted with two Gaussian distributions (solid lines). Scale bar for diffraction patterns = 2 nm⁻¹.

structures at the interfaces are quasi-intermediates formed at high-interfacial energy facets that are associated with accelerating the growth of ice hetero-nanocrystals.

In conclusion, we utilized cryo-EM and MD simulations to track the early-stage growths of individual ice nanocrystal polymorphs in an ice film of nanoscale thickness to reveal their distinct growth dynamics and interfaces in the early stage of ice crystallization. Our results revealed that $I_c$ domains are relatively small in area and exhibit limited growth, whereas hetero-crystalline, or $I_{c+h}$ domains undergo continual growth, with the growth rates along the prism planes exceeding those

along the basal planes. We found that this growth anisotropy is attributed to the distinct densities and structures of interfacial regions of growing nanocrystals. While water molecules near growing ice $I_c$ crystals do not exhibit significant differences in the density or structure compared to bulk water, the fast-growing prismatic planes of ice $I_{c+h}$ reveal the presence of a quasi-ice interface, a region that may correspond to LDL, with higher tetrahedral order and lower density than liquid water. Through this process, we fundamentally elucidated the nature and role of the interfacial properties on ice growth, further advancing the studies of nanocrystalline ice which is crucial for

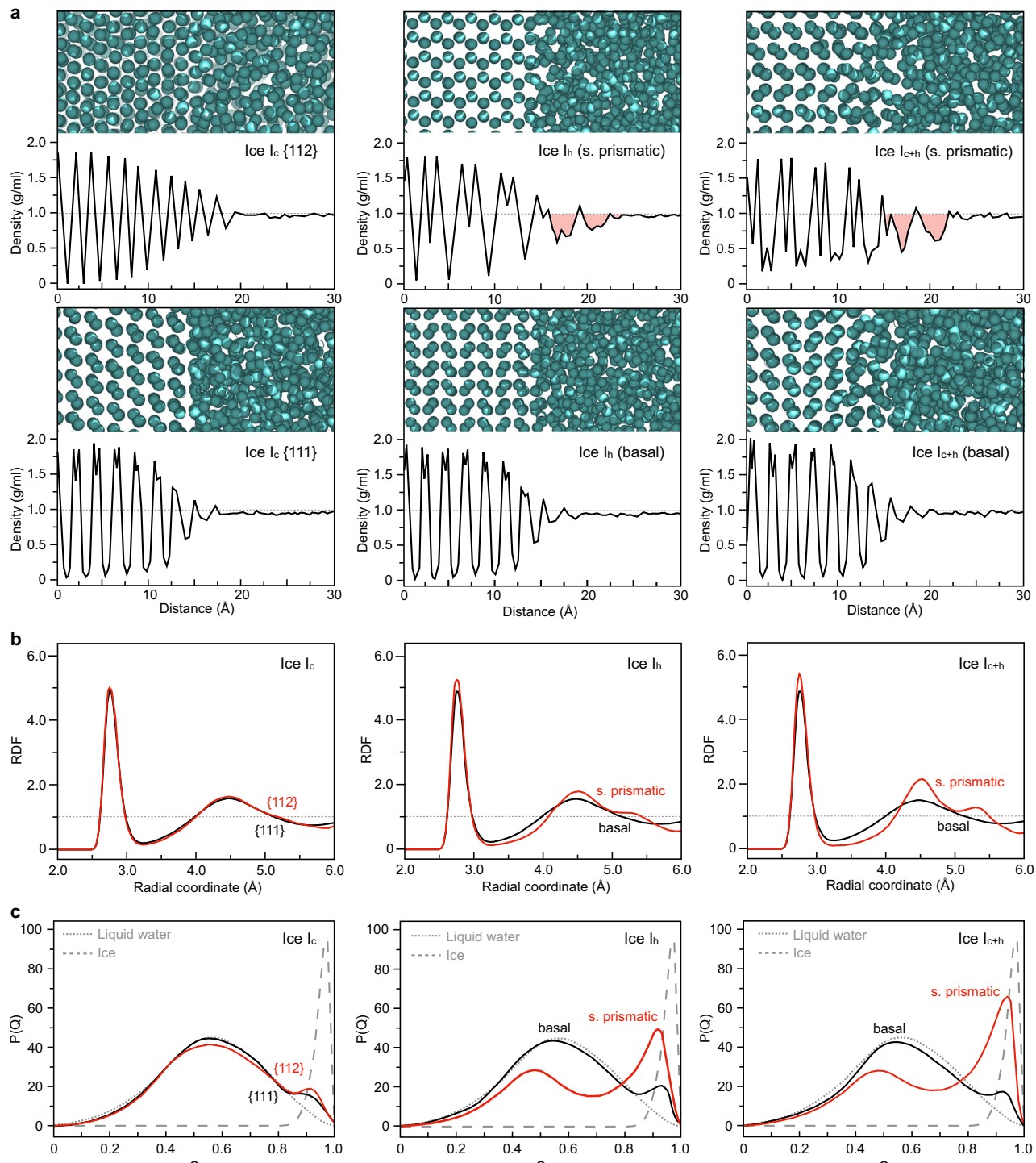

**Fig. 5 | Molecular configuration properties at solid-liquid interfaces of ice polymorphs. a** Molecular arrangement and density profiles of $H_2O$ molecules at solid-liquid interfaces in proximity to ice $I_c$ (first column), ice $I_h$ (second column), and ice $I_{c+h}$ (third column) in relation to the different ice planes, obtained from MD simulations. Low-density regions at the interface are emphasized with red shading. **b** Radial distribution function and **c** tetrahedral order parameter analysis of interfacial $H_2O$ molecules adjacent to ice $I_c$ (first column), ice $I_h$ (second column), and ice $I_{c+h}$ (third column) facets.

understanding phase transitions of ice, cloud physics, and applications such as designing various cryo-protective molecules.

## Methods
### Amorphous ice film preparation
Milli-Q Ultrapure distilled (DI) water was used to prepare the ice films. A Quantifoil™ Au 2/2 200 mesh grid was glow discharged for 30 s at 15 mA. 3 µL of DI water was pipetted onto the carbon film side of the grid, and the excess water was blotted for 1.5 s with a force of 1 before being plunge-frozen in liquid ethane using Vitrobot Mark IV. This produces nanometer-scale thin amorphous ice films suspended in the holes of the holey-carbon film. The samples were transferred and stored in liquid nitrogen for no longer than one week prior to imaging.

## Energy-filtered transmission electron microscopy for measuring ice thickness

Measurement of ice thickness was performed using energy-filtered transmission electron microscopy (EFTEM), which allows for calculation of the fraction of inelastically scattered electrons to determine sample thickness (Supplementary Fig. 1). The sample thickness is measured with Poisson statistics of inelastic scattering shown by the equation

$$\frac{t}{\lambda} = -\ln\left(\frac{I_0}{I_t}\right) \tag{2}$$

where $t$ is the sample thickness, $\lambda$ is the inelastic mean free path of the sample, and $I_0$ and $I_t$ are the zero-loss intensities and the total intensities respectively. The inelastic mean free path value for ice at 200 keV, which is 287 nm[61], was used for calculations.

## Temperature-controlled cryo-electron microscopy of ice films

Bright-field transmission electron microscopy (BFTEM), high resolution TEM (HRTEM), selected area electron diffraction (SAED), nanobeam diffraction (NBD), and z-contrast image analysis of ice were performed on a JEM-2100F (JEOL Ltd.) equipped with an UltraScan 1000XP CCD detector (Gatan) under an acceleration voltage of 200 kV. The temperature of the sample was controlled using a cryo-transfer holder (Model 626, Gatan), which allows the ice film to be ramped to and maintained at a desired temperature. We note that the electron beam was blanked during the temperature ramp.

BFTEM was performed by inserting a small objective aperture and selecting only the transmitted beam, so that electrons scattered to higher angles are excluded and less crystalline fringes appear in the images. Annealing of ice films was chosen to be performed at 143 K because crystalline structures start to occur when the temperature reaches 143 K. The dose rate used for imaging was 3.06 electrons/Å²/s, with exposures of 1.0 s. The images of holes were acquired in a sequential manner, so that annealing at different times were captured but each hole was only exposed with the electron beam once. Holes that were near the Au mesh tend to have thicker ice and were excluded from imaging. Additionally, holes with ice contaminants from vapor deposits were also excluded. Sequential TEM images of holes were acquired as soon as the temperature controller indicated that the holder temperature reached 143 K. To identify the time elapsed after annealing at 143 K, the time data for each image was extracted and compared to the time the first image was taken. The in-situ BFTEM images were performed by first allowing ice nanocrystals to form and grow on the ice with the beam blanked, then obtaining images at intervals of 11–15 s, with the electron beam blanked between consecutive frames. Each exposure was recorded at 7.5 frames per second for 10–15 frames each.

For HRTEM imaging of the ice nanocrystals, the ice film was annealed for at least 600 s until most of the amorphous ice had crystallized. Then, the cryo-transfer holder temperature was brought back down to 93 K to alleviate electron beam induced damage, which occurs more prominently at higher temperatures. SAED patterns of individual ice $I_c$ and ice $I_{c+h}$ crystals in amorphous ice were acquired by annealing amorphous ice films at 143 K for 60 s before the cryo-transfer holder temperature was brought back down to 93 K. The SA aperture with a diameter of 150 nm was placed at the location of the individual ice domains, allowing us to obtain diffraction patterns of a single domain without a second domain forming within the imaging area. NBD patterns of ice $I_{c+h}$ particles were acquired with the same methods for SAED patterns. The NBD probe size is estimated to be 105.6 nm. The z-contrast image of ice $I_{c+h}$ was obtained by using high-angle annular dark field scanning transmission electron microscopy (HAADF-STEM).

## Ice characterization with electron diffraction

Electron diffraction of individual particles were performed using the selected area (SA) aperture. The desired nanocrystals were found and selected using TEM imaging, and then the SA aperture was placed onto the position containing the desired nanocrystal. Then, the electron beam was blanked and changed to diffraction mode. The diffraction patterns were obtained as soon as the electron beam was turned back on. The position of the aperture was recorded by obtaining a post-exposure image after going back to TEM imaging mode. NBD experiments were performed in a similar way. Diffraction patterns of individual ice domains were indexed using known lattice constants of ice $I_c$ and ice $I_h$ crystal structures. Diffraction patterns of ice $I_{c+h}$ domains with streaks were compared to ED simulations performed with clTEM[62] to identify the contributions from $I_c$ and $I_h$ sequences.

Radial averages of multi-crystalline or amorphous diffraction patterns were obtained with DiffTools[63] implemented in the Gatan Software. For all the radial averages of diffraction patterns, the angular range was selected so that the beam stopper was excluded.

The powder diffraction simulations for x-ray diffraction (XRD) spectra of ice $I_c$ and $I_h$ shown in Fig. 1j were performed using VESTA software[64] to compare the presence of $I_c$ and $I_h$ sequences obtained from electron diffraction from annealed amorphous ice films. Peak positions of simulated XRD spectra in $2\theta$ (deg) were calibrated with momentum transfer values in nm⁻¹ obtained from radial averages of electron diffraction by the following procedure.

The momentum transfer $k$, in nm⁻¹ was converted to $Q$, in Å⁻¹. $k$ and $Q$ are parameters used in electron and x-ray diffraction, respectively, and their relationship is determined by the following equation derived from their relationships with the lattice spacing.

$$Q(\text{Å}^{-1}) = \frac{2\pi k(\text{nm}^{-1})}{10(\text{nm}^{-1}/\text{Å}^{-1})} \tag{3}$$

The $Q(\text{Å}^{-1})$ values calculated from this equation was used to plot radial averages of ED patterns in Figs. 1k and 4g. $2\theta$, in deg, can then be calculated by the following equation shown in terms of $Q$

$$2\theta(\text{deg}) = 2^* \sin^{-1}\left(\frac{Q\lambda}{4\pi}\right) \tag{4}$$

where $\lambda = 1.54$ Å was used for the simulations. The values of $2\theta$ were used to calibrate the top and bottom independent variable axes of Fig. 1k.

For the diffraction pattern obtained from the interfacial region of an ice $I_{c+h}$ particle and the amorphous ice film at 143 K shown in Fig. 4h, i, we radially-averaged an angular range of 100 degrees that do not contain any high-intensity crystalline peaks. Background subtraction was performed by subtracting the baseline with the Origin Pro software, and the first two peaks arising from the amorphous component were fitted with the sum of two Gaussian functions. This was performed on the amorphous diffraction pattern to compare the positions and the full width at half maximum (FWHM) of the peaks of the amorphous components.

## Image processing and nanocrystal quantification

To analyze the areas of ice domains from the images, we used the local contrast enhancement (CLAHE) algorithm and the background subtraction algorithm implemented in the ImageJ software[65]. The shape contours of the domains were obtained using Gaussian filter, thresholding and binarization, while additional adjustments or adjacent domains were delineated by manual outlining. The in-situ movies were analyzed in the same way, but each frame of the final movie was averaged amongst 6 frames to decrease background noise. Additionally, the in-situ movies were drift-corrected with a MATLAB code. Noise in image is filtered out using a Gaussian filter and feature points were

extracted using MATLAB's built-in detectSURFeatures function. We extract drift of images by calculating the shift value that minimizes the distance between feature points between two adjacent frames. The areas and diameters of the contours obtained with binarization were then measured with ImageJ. Only crystalline domains within the 2 μm diameter hole of the Quantifoil™ grid were analyzed.

## TEM and ED simulations

We simulated TEM images and ED patterns of ice $I_c$, ice $I_h$, and ice $I_{c+h}$ to validate our experimental HRTEM images. The models of $I_c$, $I_h$, and $I_{c+h}$ were generated by MD simulations described in the next section. The TEM simulations and ED simulations were performed by using GPU-accelerated multislice algorithm in clTEM[62]. The parameters including accelerating voltage, defocus, and Cs were used based on parameters for JEM-2100F: 200 kV, 5 nm, and 1 mm, respectively. Simulated noise was generated by selecting the Orius detector with a binning of 4.

## System parameters for MD simulations

AAMD simulations were performed with GROMACS 2022 simulation package[66]. Considering the realistic behavior of water molecules at low temperature, the TIP4P/ICE water molecule was used[67]. The crystal structures of cubic and hexagonal ice were adapted from the Crystallography Open Database. Regarding pressure coupling, the Berendsen barostat[68] and Parrinello-Rahman barostat[69] were used to maintain the pressure at equilibrium and production step, respectively, with coupling constants of 1.0 and 12.0 ps$^{-1}$. The temperature was kept constant at 233 K to observe the change of water molecules at low temperature. The neighbor list was built using the Verlet cut-off scheme with a cut-off of 1.2 nm at each step. The LINCS algorithm[70,71] was used to constrain the bond lengths. The leap-frog integrator with a time step of 2 fs was used. The electrostatic interactions were calculated using PME[72] with a cutoff of 1.2 nm.

## System design for three-dimensional ice growing simulations

The ice seeds were created by editing the Crystallographic Information Files of cubic, hexagonal, and hetero-crystalline ice. We initialized our system using a spherical seed with a diameter of 6 nm, serving as the nucleus for ice crystallization. The system was contained within a cubic simulation box with a side length of 20 nm to provide ample volume for subsequent ice growth. The system was equilibrated at 233 K using a v-rescale thermostat. We opted for this thermostat due to its efficient temperature control and adaptability in simulating complex molecular systems. To maintain the integrity of the ice seed during initial equilibration, position constraints were applied, ensuring that it remained intact over an equilibration period of 10 ns. After stabilization, we removed the position constraints to allow for the natural growth dynamics of the ice structure. The production phase of the MD simulation was then carried out at a consistent temperature of 233 K. This phase aimed to capture the intricate details of ice nucleation and growth, originating from the seed. The temperature of system (233 K) also allowed us to observe ice dynamics without excessive computational overhead, making the most of our available computational resources.

## Defining interfacial water at ice-water interface

The interface was delineated based on the thickness of the quasi-liquid layer (QLL) using a density profile analysis. For each simulation snapshot, we computed a one-dimensional density profile, ρ(z), perpendicular to the ice/water interface. This process entailed segmenting the simulation box into slender slabs and tallying the water molecules in each slab to determine the local density. Peaks in this density profile pinpointed the ice and liquid phases. The region with diminished density between these peaks was ascribed to the QLL. Consequently, the thickness of the QLL was defined by the span of this intermediate density region, offering a precise boundary for the ice/water interface.

## Data availability

The data supporting the findings of this study are available from the corresponding authors upon request. Source data are provided with this paper.

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

## Acknowledgements

This work was supported by the National Research Foundation of Korea (NRF-2017M3D1A1039421, 2021R1A2C3009955 (D.J.A.), 2021M3A9I4022936, 2017R1A5A1015365 (J.P.), 2020R1I1A1A0107416613 (M.-H.K.), and RS-2023-00214066 (S.Y.L.)). The authors acknowledge

financial support from the Samsung Science and Technology Foundation (SSTF-BA1802-08) for developing in-situ, cryo-TEM experiments, and data analysis methods (J.P.). The authors additionally acknowledge financial support from the Institute for Basic Science (IBS-R006-D1) (J.P.).

## Author contributions

D.J.A. and J.P. conceived the concept of this work and supervised overall experiments and simulations. M.L., S.Y.L. and M.-H.K. contributed equally to this work. M.L., S.Y.L., M.-H.K., T.K.W., J.P. and D.J.A. wrote the manuscript. M.L., M.-H.K., T.K.W. and S.K. planned the electron microscopy experiments, and S.Y.L. and T.K.W. planned the MD simulations of the research. M.L. and M.-H.K. captured the TEM images and in situ movies. M.L. conducted image analysis of the ice crystals and their interfaces. S.Y.L. conducted the MD simulations and theoretical interfacial structure analysis. S.K. and J.K. developed the analytical methods for the experimental data. All authors contributed to the discussion of results.

## Competing interests

The authors declare no competing interests.
