## [Peer Review File · Nature Communications]

Observing growth and interfacial dynamics of nanocrystalline ice in thin amorphous ice filmsReviewer #1 (Remarks to the Author):

In this work, Minyoung Lee and collaborators perform experiments (eg, in-situ electron microscopy, x-ray diffraction) and molecular dynamics simulations to study the nucleation and growth of nanoscale domains of ice at normal pressure within samples of amorphous ice. Specifically, amorphous ice films are prepared by hyper-quenching liquid water to low temperatures ($T < 100$ K) and then, the samples are annealed at $T = 143$ K at which nanoscale domains of ice [cubic ice (Ic), and heterogeneous hexagonal/cubic ice (Ih+Ic)] form and grow over time. It is found that in the early stages of the process, the nanoscale ice crystals' growth kinetics depends on the ice form (Ic and Ic+Ih). For example, small ice Ic domains exhibit finite growth while ice domains composed of both ice Ic+Ih grow continuously. In the case of heterogeneous samples composed of ice Ih+Ic, the ice growth is anisotropic with faster growth at the prismatic plane.

The phenomenon of ice nucleation is important in scientific and technological applications, for example, for the development of cloud and climate models. The topic of this work is relevant and, indeed, numerous computational/experimental studies have addressed this issue in the past. Experimentally, it is difficult to make samples of well-defined composition (ice Ic, Ih, or mixtures of ice Ic and Ih), and common diffraction techniques can only sample spatial-average properties of the samples studied. In this regard, the authors stress that these problems are avoided in their in-situ cryoEM experiments of the nanoscale ices.

The manuscript is well-written and the results are interesting. I think the manuscript may be suitable for publication in Nature Communications. However, in its present form, I cannot recommend the publication of the manuscript in Nat Comm. Below are a few points that I believe need to be addressed.

1) Unless I missed it, the thickness of the amorphous ice films studied is not given in the manuscript. Is the film thick enough to avoid any effect of the film interfaces on the growth of the nanoscale ice domains (over the time scale of the experiments, i.e., 1000 secs)?

2) In line ~110, it is stated that small ice nanocrystals with areas $< 7000 \text{ nm}^2$ are always present. The authors conclude that there are small nanocrystals that "remain small and exhibit finite growth". Can the authors discuss why the ice nucleation/growth process suddenly stops? Once the critical nucleus size is reached, the ice nucleus is expected to continue growing.

3) Ice nucleation is usually studied in the low temperature and supercooled liquid state of water, at $T > 235$ K. In this work, ice growth is studied at $T = 143$ K, in amorphous ice. This should be stressed in the manuscript. What are the expected differences, if any, in the corresponding ice growth mechanisms?

Similarly, there is a fundamental difference between the MD simulations and the exps. The ice+liquid water simulations are performed at $T = 233$ K where TIP4P/Ice is a supercooled liquid, where water molecules can easily diffuse. Instead, the exps are performed at $T = 143$ K, in the "ultraviscous liquid state" of water, just above water's glass transition temperature at which water molecules barely diffuse (over 1000 secs). The authors should stress these differences between exps and simulations and emphasize that the MD simulations are just indicative of how ice may grow at the different ice surfaces considered. This is important since the growth rate of ice Ih and Ic may vary differently with T.

4) a) In lines ~581-584, the preparation of the simulation boxes is unclear. What are the dimensions of the simulation box? What are the simulation times involved for thermalization? What thermostat is used in the MD simulations? Is there any reason for the temperature $T = 233$ K chosen for the MD simulations?

b) in the snapshots of Fig. 3j, it seems that the structure of the ice seed (ice Ih, Ic, and Ic+Ih) is conserved during the ice growth process (200 ns). However, at least for the case of stacking ice

Ic/Ih, ice Ic domain should transform, at some point, into ice Ih. In Fig. 3j, it would be important to see the molecules in the Ic and Ih domains with different colors. This may highlight any defects forming in the ice and in particular, the kind of ice that forms around the ice seed.

5) a) The interface of the ice Ic+Ih nanocrystals is discussed in lines ~235-244 and Fig. 4. The authors suggest that the density of such an interface is lower than the density of LDA. This seems surprising. LDA (low-density amorphous ice) has a very low density $\sim 0.94 \text{ g/cm}^3$, very close to the density of ice Ih ($\sim 0.92 \text{ g/cm}^3$). Can the authors estimate the density of the interface of ice Ih+Ic nanocrystals? (eg., by using the first peak position of the patterns shown in Fig. 4g). Assuming the contrast images in Fig. 4 are reliable, how sensitive are they? Can these images be used to detect differences in density of only $1-2 \text{ g/cm}^3$?

b) To support the findings discussed in (a), the authors study the interface of ice Ic and Ic+Ih using MD simulations (Fig. 5). They argue that the density of liquid water next to the studied ices is depleted next to the ice Ic+Ih interfaces (red shaded areas in Fig. 5a). However, the density profile in Fig. 5a are difficult to interpret. The minima/maxima in the profiles are due to layering of the water molecules in the liquid surrounding the ice. The authors should calculate a local coarse-grained density of water by using thick slabs ($\sim > 5 \text{ \AA}$), $\rho_{\text{ave}}(d)$. A line should also be included in these fig at the value of "d=distance" where the ice domain approximately ends (and the interfacial region begins). By comparing the $\rho_{\text{ave}}(d)$ for the ice Ic and ice Ih+Ic, one may conclude whether the density of water at the interface of ice Ih+Ic is indeed depleted.

c) The RDF and P(Q) distributions shown in Fig. 5 are also difficult to interpret and hence, the corresponding conclusions remain unclear. What molecules are used to calculate these properties? i.e., how do the authors identify the molecules belonging to the interfacial region surrounding ice Ic+Ih? How do their results depend on the definition of such "interfacial molecules"?

Reviewer #2 (Remarks to the Author):

Via in-situ cryo-electron microscopy and computational ice-dynamics simulations, the author investigated ice growth and revealed that nanoscale ice crystals exhibit polymorph-dependent growth kinetics, while hetero-crystalline ice exhibits anisotropic growth, with accelerated growth occurring at the prismatic planes. Fast-growing facets are associated with low-density interfaces that possess higher surface energy, driving tetrahedral ordering of interfacial H₂O molecules and accelerating ice growth. The authors did a detailed analysis of HRTEM images, diffraction patterns, and particle sizes, and compared them with simulation results. The results are solid and convincing, and I recommend accepting the manuscript with minor revisions.

1. Fig. 4 e and f, do the authors have high-resolution TEM images of the interfacial area?
2. The description needs to be clearer.

For example, the authors mentioned many times of density. I assume it is water density. It may be clear for people who study ice growth, but it is not clear what density the authors talk about for general readers.

Another example, Fig. 4e, I assume the dashed white line circle is the aperture, but it is not described anywhere.

Reviewer #3 (Remarks to the Author):

In this article, M. Lee et al reported the growth and interfacial dynamics of nanocrystalline ice using in-situ cryo-electron microscopy and computational ice-dynamics simulations. The authors claimed that nanoscale ice crystals exhibit polymorph-dependent growth kinetics, while hetero-crystalline ice exhibits anisotropic growth, with accelerated growth occurring at the prismatic planes. Fast-growing facets are associated with low-density interfaces that possess higher surface energy, driving tetrahedral ordering of interfacial H₂O molecules and accelerating ice growth. The direct observation of nanoscale ice crystals in situ with electron microscopy methods is very challenging, arising from the electron beam radiation, low image contrast, potential impurities in

the sample and others. It is clear that the authors in this work made a lot of effort, however, this reviewer has major concerns as listed in the following:

1). In-situ movies are not available for this review; they should be provided. Not sure whether it is the review portal issues, or missed submission.

2). In Figure 1, the amorphous structure under continuous electron beam irradiation showed nucleation and growth of ice nanocrystals at 94k. The authors also showed under lower temperature 143k and after annealing, crystalline ice with different structures were found. The slow and fast growth are not clearly distinguished. And, the electron beam effects, and the difference from the annealing are not well explained.

3). The ice crystals of different atomic structures are shown in Figure 2. The quality of the images is not good enough to directly compare with atomic positions/structures in the atomic models. It is well known that the different sample thickness can change the high resolution images significantly.

4). The image contrast of nanocrystals is very high. It is a concern that those nanoparticles can be from impurities. More analysis, such as EELS are needed to confirm it is ice. In addition, chemical mapping is also necessary to exclude other possible origins.

5). The simulated conditions are not representative of the real conditions in this in situ Cryo-EM experiments. For example, at 94k the amorphous ice was found, which is not representative of the common scenario for ice structure formation.

Response to Reviewer Remarks

Reviewer #1 (Remarks to the Author):

In this work, Minyoung Lee and collaborators perform experiments (eg, in-situ electron microscopy, x-ray diffraction] and molecular dynamics simulations to study the nucleation and growth of nanoscale domains of ice at normal pressure within samples of amorphous ice. Specifically, amorphous ice films are prepared by hyper-quenching liquid water to low temperatures ($T < 100$ K) and then, the samples are annealed at $T = 143$ K at which nanoscale domains of ice [cubic ice (Ic), and heterogeneous hexagonal/cubic ice (Ih+Ic)] form and grow over time. It is found that in the early stages of the process, the nanoscale ice crystals' growth kinetics depends on the ice form (Ic and Ic+Ih). For example, small ice Ic domains exhibit finite growth while ice domains composed of both ice Ic+Ih grow continuously. In the case of heterogeneous samples composed of ice Ih+Ic, the ice growth is anisotropic with faster growth at the prismatic plane.

The phenomenon of ice nucleation is important in scientific and technological applications, for example, for the development of cloud and climate models. The topic of this work is relevant and, indeed, numerous computational/experimental studies have addressed this issue in the past. Experimentally, it is difficult to make samples of well-defined composition (ice Ic, Ih, or mixtures of ice Ic and Ih), and common diffraction techniques can only sample spatial-average properties of the samples studied. In this regard, the authors stress that these problems are avoided in their in-situ cryoEM experiments of the nanoscale ices.

The manuscript is well-written and the results are interesting. I think the manuscript may be suitable for publication in Nature Communications. However, in its present form, I cannot recommend the publication of the manuscript in Nat Comm. Below are a few points that I believe need to be addressed.

Response to the general comment:

We highly appreciate Reviewer #1's valuable and helpful comments. We have done our best to provide appropriate responses and detailed experimental processes in the letter and the revised manuscript to address these questions.

Comment #1:

1) Unless I missed it, the thickness of the amorphous ice films studied is not given in the manuscript. Is the film thick enough to avoid any effect of the film interfaces on the growth of the nanoscale ice domains (over the time scale of the experiments, i.e., 1000 secs)?

Response #1:

We appreciate the reviewers for bringing up this important comment on the effects of the film interface. To measure the thickness of ice films, we have performed energy-filtered transmission electron microscopy (EFTEM) shown in the added Supplementary Fig. 1, which allows for calculation of the fraction of inelastically scattered electrons to determine sample thickness. The sample thickness is measured by the equation $\frac{t}{\lambda} = -\ln\left(\frac{I_0}{I_t}\right)$ where t is the sample

thickness, λ is the inelastic mean free path of the sample, and I_0 and I_t are the zero-loss intensities and the total intensities respectively. Using the inelastic mean free path value for ice at 200 keV, which is 287 nm¹, the mean ice thickness measured is 132 nm with a standard deviation of 13.4 nm. Amorphous ice films deposited on Pt(111) is known to exhibit thickness dependence in crystallization kinetics at thicknesses up to 30 nm due to the large surface area of interfaces, which accelerate nucleation and crystallization²⁻⁴. Beyond around 75 nm, the thickness dependence is reported to be minimal⁵.

While our amorphous ice film is estimated to be at a scale thick enough to consider the nucleation and crystallization as a bulk process, we note that once an ice domain's diameter exceeds the thickness of the amorphous ice (i.e. 132 nm), growth along the thickness axis of the ice film stops, as illustrated by a scheme added as Supplementary Fig. 8a. In our experiment, this starts to occur after 215 s of annealing as shown in Supplementary Fig. 8b, where ice nanocrystal diameters start to increase rapidly beyond around 300 nm, becoming the dominant mechanism of crystallization. After 215 s, the fitted Avrami parameter becomes $n = 2.98$, based on the log-log plot in Supplementary Fig. 8c, which indicates that nanocrystals start to exhibit growth along two dimensions effectively⁶. Based on these results, we can conclude that, during crystallization, nanocrystals first nucleate within amorphous ice, followed by their growth until around 132 nm, and if their diameters reach beyond 132 nm they would grow large enough to impinge on the surface and their growths along the thickness axis will be halted.

Despite the limited growth of ice domains along the thickness axis for larger ice nanocrystals, our main conclusions do not change regarding the accelerated growth of the secondary prismatic plane of ice I_{c+h} and the formation of the low-density crystalline-amorphous growth interface structures. *In-situ* measurements on heterocrystalline ice with initial diameters of 88.9, 128.8, and 302.4 nm (Fig. 3c and Supplementary Fig. 14) all exhibit accelerated growth along the secondary prism plane growth direction, which shows that the dynamics we have observed are unchanged among different sizes of ice nanocrystals. In addition, for these ice domains, the directions of the basal plane growths and the secondary prismatic plane growths are normal to the thickness direction and are both directed along the bulk amorphous ice.

In response to the reviewer's comment, we have added a discussion in the main text on the effect of the interface that causes growth along two dimensions for larger ice domains.

Added Figure (Supplementary information, page 7, lines 127-133)

Supplementary Fig. 1: EFTEM analysis for amorphous ice thickness measurement.

a EFTEM image of a carbon film hole with a free-standing amorphous ice film in the region inside the hole. Each pixel is color-mapped based on t/λ values. The thickness t is calculated based on the inelastic mean free path (IMFP) value of ice 287 nm at 200 keV (Scale bar = 200 nm). **b** Histogram of ice thickness values obtained for each pixel located inside holes, taken from multiple holes.

Added Figure (Supplementary information, page 14, lines 180-189)

Supplementary Fig. 8: Analysis for growth modes of ice domains according to different sizes.

a Schematic of the proposed growth mechanisms of ice nanocrystals of different sizes. **b** Distributions of Feret diameter of ice nanocrystals according to annealing time in between 0 s and 402 s, where we identify the point in time in which particle sizes undergo a sudden increase in diameter, which occurs after 215 s of annealing. **c** Log-log graph of the Avrami equation, which was calculated with the crystallized fraction and the annealing time. The graph is fit with two separate regimes of growth delineated by the yellow dotted line, fitted with $n = 0.613$ and $n = 2.98$ (red lines).

Added text (Main text, page 4, lines 80-82)

The average thickness of the free-standing amorphous ice, determined by energy filtered TEM, is 132 nm with a standard deviation of 13.4 nm (Supplementary Fig. 1), whereby crystallization kinetics is not thickness-dependent⁴⁶⁻⁴⁹.

Added text (Main text, page 5, lines 111-119)

The overall Avrami parameter n is close to 3, indicating that the system effectively exhibits growth along two dimensions⁵⁰. This is because ice nanoparticles with diameters exceeding around 132 nm, the thickness of the ice film, are expected to stop growing along the thickness axis but continue growth along the directions normal to the thickness (Supplementary Fig. 8a). This type of growth is expected to be especially prominent after annealing for more than 215 s, where ice nanocrystal diameters exhibit a sudden increase (Supplementary Fig. 8b). We note that the Avrami parameter at this regime after 215 s was fitted for $n = 2.98$, as shown in Supplementary Fig. 8c.

Added text (Main text, Methods, page 23, lines 411-418)

Energy-filtered transmission electron microscopy for measuring ice thickness

Measurement of ice thickness was performed using energy-filtered transmission electron microscopy (EFTEM), which allows for calculation of the fraction of inelastically scattered electrons to determine sample thickness (Supplementary Fig. 1). The sample thickness is measured with Poisson statistics of inelastic scattering shown by the equation $\frac{t}{\lambda} = -\ln\left(\frac{I_0}{I_t}\right)$ where t is the sample thickness, λ is the inelastic mean free path of the sample, and I_0 and I_t are the zero-loss intensities and the total intensities respectively. The inelastic mean free path value for ice at 200 keV, which is 287 nm⁶¹, was used for calculations.

Comment #2:

2) In line ~110, it is stated that small ice nanocrystals with areas <7000 nm² are always present. The authors conclude that there are small nanocrystals that “remain small and exhibit finite growth”. Can the authors discuss why the ice nucleation/growth process suddenly stops? Once the critical nucleus size is reached, the ice nucleus is expected to continue growing.

Response #2:

Our observations show that the nucleation/growth process of small I_c nanocrystals do not stop completely but rather grow at a much slower rate than the I_{c+h} crystals. We have provided the graph of changes in area over time for the small ice nanocrystals shown in Fig. 3e and f with adjusted scale (Figure R1a and b, respectively), which show that ice nanocrystals still increase in size, although at a smaller extent.

The difference in growth rates between I_c and I_{c+h} can be attributed to the relatively lower interfacial energy we have calculated (22.1 mJ/m²) of cubic ice, as well as the limited diffusion of amorphous ice molecules contribute to slowing down the growth of the crystal at this

particular size. In the case for the fast-growing ice I_{c+h} domains, we have calculated a higher interfacial energy for the crystalline-amorphous growth interface of the secondary prismatic plane (25.4 mJ/m^2), which would drive molecular ordering at the surroundings to minimize the high interfacial energy. As a result, we observe I_{c+h} domains that continually increase in size at the expense of water molecules in the surrounding amorphous ice that become visibly depleted.

We have made revisions to the manuscript to clarify that the I_c nanocrystal growth does not stop, but is slowed down compared to the case for I_{c+h} nanocrystals, which are larger and fast-growing.

Figure R1, for reviewers only:

Figure R1. Growth of ice I_c nanocrystals.

a Area vs. time graph and **b** Areal growth rate vs. time distribution of ice I_c nanocrystals shown in Figs. 3e and f in the Main text, respectively, with enlarged y-axis scale. Ice I_c crystals continue growth throughout the time of our experiment.

Original text (Main text, page 5, lines 103-104)

...suggesting the presence of a type of domain that remains small and exhibits finite growth.

Revised text (Main text, page 6, lines 127-128)

... suggesting the presence of a type of domain that remains small with relatively slowed growth.

Original text (Main text, page 5, lines 108-110)

... These results verify that one type of domain remains a small size and does not continuously grow into large domains, while another type of domain keeps growing to a larger size until the amorphous ice is fully crystallized.

Revised text (Main text, page 6, lines 132-134)

... These results verify that one type of domain remains a small size **due to slow growth** and does not continuously grow into large domains, while another type of domain keeps growing to a larger size until the amorphous ice is fully crystallized.

Original text (Main text, page 6, lines 130-132)

The relatively high intensity of the peak at 1.65 \AA^{-1} compared to its shoulder peaks indicate that larger, continuously-growing ice domains consist of both I_c and I_h sequences and are hetero-nanocrystalline (namely, ice I_{c+h}).

Revised text (Main text, page 7, lines 154-156)

The relatively high intensity of the peak at 1.65 \AA^{-1} compared to its shoulder peaks ~~indicate~~ **indicates that fast-growing larger** ice domains consist of both I_c and I_h sequences and are hetero-nanocrystalline (namely, ice I_{c+h}).

Original text (Main text, page 7, lines 163-164)

Ice I_c particles labelled from P1 to P5 in Fig. 3a do not exhibit significant growth over the observation period, as shown by the plot of the change in area over time in Fig. 3b.

Revised text (Main text, page 9, lines 194-195)

Ice I_c particles labelled from P1 to P5 in Fig. 3a exhibit **relatively slow** growth over the observation period, as shown by the plot of the change in area over time in Fig. 3b.

Original text (Main text, page 8, lines 174-175)

Tracking the areas of 65 crystalline domains over time, we verified the dynamics of the limited and fast growth propensities of ice I_c and ice I_{c+h} , respectively.

Revised text (Main text, page 9, lines 204-205)

Tracking the areas of 65 crystalline domains over time, we verified the dynamics of the **slow** and fast growth propensities of ice I_c and ice I_{c+h} , respectively.

Comment #3:

3) Ice nucleation is usually studied in the low temperature and supercooled liquid state of water, at $T > 235 \text{ K}$. In this work, ice growth is studied at $T = 143 \text{ K}$, in amorphous ice. This should be stressed in the manuscript. What are the expected differences, if any, in the corresponding ice growth mechanisms?

Similarly, there is a fundamental difference between the MD simulations and the exps. The ice+liquid water simulations are performed at $T = 233 \text{ K}$ where TIP4P/Ice is a supercooled liquid, where water molecules can easily diffuse. Instead, the exps are performed at $T = 143 \text{ K}$, in the

“ultraviscous liquid state” of water, just above water’s glass transition temperature at which water molecules barely diffuse (over 1000 secs). The authors should stress these differences between exps and simulations and emphasize that the MD simulations are just indicative of how ice may grow at the different ice surfaces considered. This is important since the growth rate of ice I_h and I_c may vary differently with T.

Response #3:

According to the reviewer's comment, water at 143 K, which is the experimental condition, is in an ultraviscous liquid state and has movements that are too slow to undergo phase transitions. In Figure R2, we observed that the TIP4P/ICE model exhibits extremely subtle movements and a very slow ice growth rate around 153 K. To observe the growth characteristics of the water-ice interface with limited computational resources up to a few microseconds, a temperature above 213 K is required. Therefore, we agree with the comment that our simulations are indicative of how ice may grow at the different ice surfaces. We have emphasized this point in the revised manuscript and provided further explanation regarding the temperature discrepancy between the simulations and the experiment.

Figure R2, for reviewers only:

Figure R2: Simulation of the ice-water system from 133 K to 233 K.

a Diffusion coefficient of H₂O molecules in bulk water. **b** Ice growth rates on different ice surfaces: ice I_h (secondary prismatic) represented by the blue line, ice I_c {112} represented by the orange line, and the interface of ice I_h and ice I_c (basal) represented by the yellow line.

Original text (Main text, page 3, lines 69-72)

In this report, we use *in-situ* cryo-electron microscopy (cryo-EM) and molecular dynamics (MD) simulations to directly observe the time-resolved growth of individual ice nanocrystal polymorphs and investigate their respective interfacial structural dynamics.

Revised text (Main text, page 3, lines 67-70)

In this report, we use *in-situ* cryo-electron microscopy (cryo-EM) and molecular dynamics

(MD) simulations for modelling growths at different ice surfaces to directly observe the time-resolved growth of individual ice nanocrystal polymorphs and investigate their respective interfacial structural dynamics.

Added text (Main text, page 10, lines 225-231)

The simulations were conducted at 233 K, under conditions where ice seeds with high curvature can grow, and water molecules possess adequate mobility for phase transitions. We note that higher temperatures increase the extent of water molecule diffusion and consequently the ice growth rate, and therefore, a higher growth rate of the ice crystal is expected compared to our experimental conditions (143 K). The simulations were modeled and interpreted as indications of how different ice surfaces undergo crystallization.

Comment #4:

4) a) In lines ~581-584, the preparation of the simulation boxes is unclear. What are the dimensions of the simulation box? What are the simulation times involved for thermalization? What thermostat is used in the MD simulations? Is there any reason for the temperature $T=233$ K chosen for the MD simulations?

Response #4:

We appreciate the reviewer for this careful note. We utilized a spherical seed with a 6 nm diameter to promote three-dimensional growth of ice at 233 K. A cubic box with a side length of 20 nm was employed to ensure sufficient volume for ice growing from the seed. The initial ice seed was stabilized over a period of 10 ns at 233 K using a v-rescale thermostat with position constraints. Subsequently, the position constraints were removed, and a production run was conducted at the same temperature. We chose 233 K because, at this temperature, a three-dimensional ice structure of a manageable size can effectively demonstrate ice growth within the confines of our computational resources. We have added more detailed information regarding the simulation in the Methods section of the revised manuscript.

Added text (Main text, Methods, page 28, line 523-537)

System design for three-dimensional ice growing simulations

The ice seeds were created by editing the Crystallographic Information Files of cubic, hexagonal, and hetero-crystalline ice. We initialized our system using a spherical seed with a diameter of 6 nm, serving as the nucleus for ice crystallization. The system was contained within a cubic simulation box with a side length of 20 nm to provide ample volume for subsequent ice growth. The system was equilibrated at 233 K using a v-rescale thermostat. We opted for this thermostat due to its efficient temperature control and adaptability in simulating complex molecular systems. To maintain the integrity of the ice seed during initial equilibration, position constraints were applied, ensuring that it remained intact over an equilibration period of 10 ns. After stabilization, we removed the position constraints to allow for the natural growth

dynamics of the ice structure. The production phase of the MD simulation was then carried out at a consistent temperature of 233 K. This phase aimed to capture the intricate details of ice nucleation and growth, originating from the seed. The temperature of system (233 K) also allowed us to observe ice dynamics without excessive computational overhead, making the most of our available computational resources.

Comment #5:

b) in the snapshots of Fig. 3j, it seems that the structure of the ice seed (ice I_h, I_c, and I_c+I_h) is conserved during the ice growth process (200 ns). However, at least for the case of stacking ice I_c/I_h, ice I_c domain should transform, at some point, into ice I_h. In Fig. 3j, it would be important to see the molecules in the I_c and I_h domains with different colors. This may highlight any defects forming in the ice and in particular, the kind of ice that forms around the ice seed.

Response #5:

We have utilized the CHILL+ algorithm⁷ to distinguish the domains of cubic ice and hexagonal ice with different colors. Ice grown from a seed exhibited differences in its crystal structure depending on the direction of growth. In prismatic plane growth (along the {110} direction), the ice mostly grew according to the seed's crystalline structure, while in the basal plane growth (along the {111} direction), the ice grew in a stacking disordered structure (Figure R3). Additionally, the phase transition of ice I_c and ice I_{c+h} seeds into Ice I_h were not observed. However, a crystallization competition between the two crystal structures could be observed by the defect formed between the sequences of ice I_c and ice I_h. The complete phase transition from cubic sequences to hexagonal sequences is prominently observed in both the monatomic water model (mW) and the machine learning coarse-grained water model (ML-BOP). The all-atomic water model we used (TIP4P/ICE) has computational limitations in simulating the phase transition, and only local competitions in the crystal structure can be observed.

In response to the reviewer's comments, we have highlighted the cubic sequences and hexagonal sequences with different colors in the revised Fig. 3j.

Figure R3, for reviewers only:

Figure R3. MD simulation of ice growth from different seeds.

Time-series images of growing simulation using spheric seeds of **a** ice I_c , **b** ice I_h , and **c** ice I_{c+h} . Using the CHILL+ algorithm, cubic ice is represented in orange, hexagonal ice is represented in blue, and other water molecules are represented in gray.

Original Figure (Main text, Figure 3, page 19, lines 368-381)

Fig. 3: Growth dynamics of ice nanocrystal polymorphs. **a**, Time-series images of representative ice I_c nanocrystals exhibiting limited growth (Scale bars = 50 nm) and **b**, changes in their areas. **c**, A continuously-growing ice I_{c+h} nanocrystal with planar defects and **d**, the change in its area (empty circles) and lengths (solid circles) along the growth directions of basal and prismatic planes. **e**, Changes in area plotted over time for 65 ice I_c (orange) and ice I_{c+h} (dark blue) nanocrystals. **f,g**, Growth rates over time for ice I_c (**f**) and ice I_{c+h} (**g**). **h,i**, Histogram of aspect ratios summed over time for ice I_c (**h**), and ice I_{c+h} (**i**). **j**, Simulated ice growth images from spherical seeds of ice I_c (first column), ice I_h (second column), and ice I_{c+h} (third column) calculated for 200 ns at a temperature of 230 K. The first and second rows display top view images along the $[111]$ or basal axis and side view images perpendicular to the axes, respectively. Oxygen atoms are represented by teal-colored spheres. The initial position of ice seed with 5 nm diameter is marked by a red dashed line. Scale bar = 2 nm. **k**, Temporal tracking

of ice plane positions for ice I_c (yellow), ice I_h (green), and ice I_{c+h} (dark blue).

Revised Figure (Main text, Figure 3, page 19, lines 368-383)

Fig. 3: Growth dynamics of ice nanocrystal polymorphs. **a**, Time-series images of representative ice I_c nanocrystals exhibiting limited growth (Scale bars = 50 nm) and **b**, changes in their areas. **c**, A continuously-growing ice I_{c+h} nanocrystal with planar defects and **d**, the change in its area (empty circles) and lengths (solid circles) along the growth directions of basal and prismatic planes. **e**, Changes in area plotted over time for 65 ice I_c (orange) and ice I_{c+h} (dark blue) nanocrystals. **f,g**, Growth rates over time for ice I_c (**f**) and ice I_{c+h} (**g**). **h,i**, Histogram of aspect ratios summed over time for ice I_c (**h**), and ice I_{c+h} (**i**). **j**, **Cross-section images of ice**

growth simulation from spherical seeds of ice I_c (first column), ice I_h (second column), and ice I_{c+h} (third column) calculated for 200 ns at a temperature of 230 K. The first and second rows display top view images along the [111] or basal axis and side view images perpendicular to the axes, respectively. Oxygen atoms in water molecules are represented by the following: cubic structure in orange, hexagonal structure in dark blue, and other water molecules as gray spheres. The initial position of ice seed with 5 nm diameter is marked by a yellow dashed line. Scale bar = 2 nm. **k**, Temporal tracking of ice plane positions for ice I_c (yellow), ice I_h (green), and ice I_{c+h} (dark blue).

Comment #6:

5) a) The interface of the ice I_c+I_h nanocrystals is discussed in lines ~235-244 and Fig. 4. The authors suggest that the density of such an interface is lower than the density of LDA. This seems surprising. LDA (low-density amorphous ice) has a very low density ~ 0.94 g/cm, very close to the density of ice I_h (~ 0.92 g/cm³). Can the authors estimate the density of the interface of ice I_h+I_c nanocrystals? (eg., by using the first peak position of the patterns shown in Fig. 4g).

Assuming the contrast images in Fig. 4 are reliable, how sensitive are they? Can these images be used to detect differences in density of only 1-2 g/cm³?

Response #6:

We thank the reviewer for bringing up this important point. To estimate the density of the interface of ice I_{c+h} nanocrystals as the reviewer pointed out, we have first combined the radial averages of electron diffraction patterns of (1) amorphous ice at 93 K, (2) amorphous ice at 143 K, (3) the interfacial area of I_{c+h} , and (4) crystalline ice into a single graph presented in the added Supplementary Fig. 18. Through the comparisons of the first peak positions of the amorphous ices, we can conclude that for (1) amorphous ice at 93 K, the peak position at 1.65 \AA^{-1} is consistent to the values for LDA observed in literature^{8,9}; for (2) bulk amorphous ice at 143 K, the first peak shifts to 1.77 \AA^{-1} , in which we can interpret as ice being denser than LDA, attributed to a transition that would be manifested by the increase of observed density as the amorphous ice transforms into a viscous form above a temperature of 136 K^{10,11}; for (3) the interfacial area of I_{c+h} the first peak is at 1.69 \AA^{-1} , which is within error range of LDA. Through these results, we clarify that for Fig. 4g in the Main text, the density of the I_{c+h} interface is consistent with LDA.

In response to the reviewer's question on the sensitivity of the TEM mode for detecting differences in density, we have performed TEM simulations on 133.4 nm-thick slabs for two representative systems — MD simulated water at 300 K (1.0 g/ml) and LDA at 100 K (0.94 g/ml). The TEM simulations were performed for different defocus values to evaluate the changes in average pixel values to determine how density affects contrast in images. Figure R4 shows that LDA exhibits a lighter contrast (higher pixel value) for five different defocus values ranging from -20 to 20 nm.

While we can theoretically claim that TEM contrast is sensitive enough to detect differences

of density between water at 300 K (1.0 g/ml) and LDA at 100 K (0.94 g/ml), in the real TEM images we have acquired, there are multiple possible origins other than the difference in density that contribute to contrast difference in the TEM images observed at the interfacial region, namely variations in thickness and the crystallinity of the ice. Nevertheless, the light contrast at the interfacial area we have observed with BFTEM have provided important clues and motives for us to investigate the crystalline-amorphous interface of fast-growing ice, because these bright contrast interfaces are resultant features of fast growing ice domains where we also expect ice thinning due to the depletion of water molecules near the growing crystal, inspiring the investigation of the actively crystallizing interfaces that accelerate crystal growth. Furthermore, we emphasize that our conclusions on ice density at the interface are mainly based on our electron diffraction results in Fig. 4g of the Main text, which provides the most reliable evaluations on the density of amorphous materials by the measurement of peak shifts observed by extracting the amorphous components and excluding the effects from thickness or crystallinity.

We have revised our manuscript to include our insights on the density of the ice I_{c+h} interface, clarify our contrast interpretations using TEM, and emphasize that electron diffraction has been used to probe ice density.

Added figure (Supplementary information, page 24, lines 257-261)

Supplementary Fig. 18 Radial averages of electron diffraction patterns.

Radial averages of SAED patterns of amorphous ice at 93 K (red), amorphous ice at 143 K (yellow), the ice I_{c+h} domain interface (light blue) and crystalline ice after annealing (dark blue). Peak positions of the first peak are marked with a horizontal line for comparison.

Figure R4, for reviewers only:

Figure R4. Estimation of the contrast difference in TEM for the different densities of LDA and water using TEM simulations.

Slabs 133.4 nm-thick of LDA at 100 K and water at 300 K, with densities of 0.94 and 1.0 g/cm³, respectively, were used for the TEM simulations. Simulations were performed for different defocus values, and the average pixel intensities of the resulting images were calculated.

Original text (Main text, page 10, lines 231-236)

This demonstrates that there are significant differences in the thickness, densities, or structures of the surroundings and the growth interface, which are associated with the slow and fast growth rates of ice I_c and ice I_{c+h} ice domains, respectively. We obtained a SAED pattern from the interfacial region of a representative ice I_{c+h} particle shown in Fig. 4e, placing the 150-nm SA aperture at a region containing a growing ice I_{c+h} particle and the interfacial region surrounding it.

Revised text (Main text, page 12, lines 267-276)

This demonstrates that there are significant differences in the thickness, ice densities, or structures of the surroundings and the growth interface, which are associated with the slow and fast growth rates of ice I_c and ice I_{c+h} ice domains, respectively.

Electron diffraction, a method that can be used to probe ice density⁵³, was performed to investigate the structure of the surrounding ice at the interfacial region of a fast-growing ice nanocrystal, which is otherwise difficult to interpret with TEM images alone (Supplementary Fig. 17 and Supplementary text 1.5). We obtained a SAED pattern from the interfacial region of a representative ice I_{c+h} particle shown in Fig. 4e, placing the 150-nm SA aperture at a region containing a growing ice I_{c+h} particle and the interfacial region surrounding it.

Original text (Main text, pages 10-11, lines 241-246)

The first peak of the SAED pattern from the ice I_{c+h} growth interface exhibits a shift to lower reciprocal distance compared to the amorphous phase at 143 K from 1.77 to 1.69 Å⁻¹ (Fig. 4g). Such shift is analogous to previous results of the transition from high-density amorphous (HDA) to low density amorphous (LDA) ice^{10,49,50}, suggesting that the amorphous region near a growing ice I_{c+h} crystal is lower in density compared to amorphous ice at 143 K.

Revised text (Main text, pages 12-13, lines 281-290)

The first peak of the SAED pattern from amorphous ice at 143 K is at 1.77 \AA^{-1} , which is shifted to a higher reciprocal distance compared to the 93 K low-density amorphous ice (LDA) peak at 1.65 \AA^{-1} (Fig. 1g). This is attributed to a transition that would be manifested by the increase of observed density of the amorphous ice before undergoing crystallization, consistent to some reports that mention the transformation of amorphous ice into a viscous form above 136 K^{54,55}. Meanwhile, the first peak from the ice I_{c+h} growth interface exhibits a shift to lower reciprocal distance compared to the amorphous phase at 143 K from 1.77 to 1.69 \AA^{-1} (Fig. 4g and Supplementary Fig. 18). Such shift is indicative of a decrease in the density of ice^{56,57}, suggesting that the amorphous region near a growing ice I_{c+h} crystal is lower in density compared to amorphous ice at 143 K.

Comment #7:

b) To support the findings discussed in (a), the authors study the interface of ice I_c and I_c+I_h using MD simulations (Fig. 5). They argue that the density of liquid water next to the studied ices is depleted next to the ice I_c+I_h interfaces (red shaded areas in Fig. 5a). However, the density profile in Fig. 5a are difficult to interpret. The minima/maxima in the profiles are due to layering of the water molecules in the liquid surrounding the ice. The authors should calculate a local coarse-grained density of water by using thick slabs ($\sim 5 \text{ \AA}$), $\rho_{\text{ave}}(d)$. A line should also be included in these fig at the value of “ $d=\text{distance}$ ” where the ice domain approximately ends (and the interfacial region begins). By comparing the $\rho_{\text{ave}}(d)$ for the ice I_c and ice I_h+I_c , one may conclude whether the density of water at the interface of ice I_h+I_c is indeed depleted.

Response #7:

In response to the reviewer's comment, we obtained profiles using a thicker density slab (5 \AA). In added Supplementary Fig. 19, the ice and water regions showed densities of 0.94 g/ml and 0.98 g/ml , respectively. For water molecules at the interface, a low-density region appeared with densities ranging between 0.92 and 0.94 g/ml . This seems to be a reduction in density caused by the transient appearance of LDL (or LDA). We have added the results of the bulk density profile to the revised supplementary information.

Added figure (Supplementary information, page 25, lines 262-266)

Supplementary Fig. 19: Local coarse-grained density profile at water-ice interface.

Bulk density profile of water molecules at interface around the ice I_c {112} (left panel), the ice I_h (s. prismatic) (center panel), and the ice I_{c+h} (s. prismatic) (right panel). The area shaded in blue represents the ice region. A 5-angstrom slab was used.

Added text (Main text, page 13, lines 296-301)

...the density profiles of water molecules in the freezing direction were calculated for ice I_c , ice I_h , and ice I_{c+h} . Low-density regions with less than 1 g/ml were observed at the interface of prismatic planes of ice I_h and ice I_{c+h} . In the coarse-grained density profile using a thick slab (5Å), which excludes the density peaks and valleys caused by the ice lattice, these low-density regions were observed (Supplementary Fig. 19).

Comment #8:

c) The RDF and P(Q) distributions shown in Fig. 5 are also difficult to interpret and hence, the corresponding conclusions remain unclear. What molecules are used to calculate these properties? i.e., how do the authors identify the molecules belonging to the interfacial region surrounding ice I_{c+h} ? How do their results depend on the definition of such “interfacial molecules”?

Response #8:

The RDF in Fig. 5 represents the results for water molecules and was calculated targeting oxygen atoms. To identify the water molecules present at the interface, we examined the thickness of the QLL in the density profile. We defined the QLL as the area between peaks in the water molecule density profile that deviates from the density pattern of water and ice, and calculated the RDF targeting water molecules in that area. To supplement the explanation of these analytical theories, we have revised the manuscript and added detailed experimental processes in the method section as follows.

Added text (Main text, Methods, pages 28-29, line 538-546)

Defining interfacial water at ice-water interface

The interface was delineated based on the thickness of the quasi-liquid layer (QLL) using a density profile analysis. For each simulation snapshot, we computed a one-dimensional density profile, $\rho(z)$, perpendicular to the ice/water interface. This process entailed segmenting the simulation box into slender slabs and tallying the water molecules in each slab to determine the local density. Peaks in this density profile pinpointed the ice and liquid phases. The region with diminished density between these peaks was ascribed to the QLL. Consequently, the thickness of the QLL was defined by the span of this intermediate density region, offering a precise boundary for the ice/water interface.

Reviewer #2 (Remarks to the Author):

Via in-situ cryo-electron microscopy and computational ice-dynamics simulations, the author investigated ice growth and revealed that nanoscale ice crystals exhibit polymorph-dependent growth kinetics, while hetero-crystalline ice exhibits anisotropic growth, with accelerated growth occurring at the prismatic planes. Fast-growing facets are associated with low-density interfaces that possess higher surface energy, driving tetrahedral ordering of interfacial H₂O molecules and accelerating ice growth. The authors did a detailed analysis of HRTEM images, diffraction patterns, and particle sizes, and compared them with simulation results. The results are solid and convincing, and I recommend accepting the manuscript with minor revisions.

Response to the general comment:

We thank Reviewer #2 for the helpful comments and suggestions. We have done our best to provide additional data and implement appropriate revisions as the reviewer suggests.

Comment #1:

1. Fig. 4 e and f, do the authors have high-resolution TEM images of the interfacial area?

Response #1:

We thank the reviewer for the great question. We have provided images of the interfacial area with HRTEM and a false-colored, low-pass, gaussian filtered version of the interfacial area as shown in the added Supplementary Fig. 17a and b, respectively. We have labelled the cubic sequences and the stacking fault visible in the HRTEM image at the crystalline region. For crystalline regions of the ice, the well-aligned atomic columns enable visibility of the atomic structures. The atomic positions are not clearly defined at the interfacial region, which is thick (around 100 nm) and amorphous in nature. According to our TEM simulation results with water molecules having a density of 1.0 g/cm³ (Supplementary Fig. 17c), the resulting simulated images obtained with a perfect detector free of noise and a CCD detector both do not allow us to specify the exact position of the oxygen atoms (Supplementary Fig. 17d). Therefore, this motivated us to use the electron diffraction method to characterize the density of the interfacial region, because the peak shifts can accurately identify the changes according to density.

Added Figure (Supplementary information, page 23, lines 250-257)

Supplementary Fig. 17: HRTEM image of the interfacial area.

a HRTEM image of an interfacial area of a crystalline ice particle. **b** False color close-up image of the region marked as the red, dotted box in **a** (Scale bars = 2 nm). The image has been low-pass filtered, and then a gaussian filter has been implemented. **c** Water model with density 1.0 g/cm^3 obtained with MD simulations. **d** Simulated TEM images of water molecules with 58 \AA thickness. Even with the perfect detector, which is free of noise, the positions of water molecules are not defined.

Added text (Main text, page 12, lines 271-274)

Electron diffraction, a method that can be used to probe ice density⁵³ was performed to investigate the structure of the surrounding ice at the interfacial region of a fast-growing ice nanocrystal, which is otherwise difficult to interpret with TEM images alone (Supplementary Fig. 17 and Supplementary text 1.4).

Added text (Supplementary information, Supplementary text, page 6, lines 117-127)

A subsection in the supplementary text titled 1.4 HRTEM image of the interfacial area was added.

Comment #2:

2. The description needs to be clearer.

For example, the authors mentioned many times of density. I assume it is water density. It may be clear for people who study ice growth, but it is not clear what density the authors talk about

for general readers.

Another example, Fig. 4e, I assume the dashed white line circle is the aperture, but it is not described anywhere.

Response #2:

We appreciate the reviewer for providing great suggestions to clarify our writing. We have made the following revisions to our Main text accordingly, to improve the clarity of our manuscript.

Original text (Main text, pages 3-4, lines 74-77)

We further elucidate that fast-growing facets are associated with low-density interfaces that possess higher surface energy, driving tetrahedral ordering of interfacial H₂O molecules and hence accelerating ice growth.

Revised text (Main text, page 4, lines 73-75)

We further elucidate that fast-growing facets are associated with a lower density of H₂O molecules at interfaces which possess higher surface energy, driving tetrahedral ordering of interfacial H₂O molecules and hence accelerating ice growth.

Original text (Main text, page 10, lines 231-234)

This demonstrates that there are significant differences in the thickness, densities, or structures of the surroundings and the growth interface, which are associated with the slow and fast growth rates of ice I_c and ice I_{c+h} ice domains, respectively.

Revised text (Main text, page 12, lines 267-270)

This demonstrates that there are significant differences in the thickness, ice densities, or structures of the surroundings and the growth interface, which are associated with the slow and fast growth rates of ice I_c and ice I_{c+h} ice domains, respectively.

Added text (Main text, Fig. 4 caption, page 21, lines 390-391)

The white dashed circle represents the area where the aperture was inserted for SAED analysis.

Reviewer #3 (Remarks to the Author):

In this article, M. Lee et al reported the growth and interfacial dynamics of nanocrystalline ice using in-situ cryo-electron microscopy and computational ice-dynamics simulations. The authors claimed that nanoscale ice crystals exhibit polymorph-dependent growth kinetics, while hetero-crystalline ice exhibits anisotropic growth, with accelerated growth occurring at the prismatic planes. Fast-growing facets are associated with low-density interfaces that possess higher surface energy, driving tetrahedral ordering of interfacial H₂O molecules and accelerating ice growth.

The direct observation of nanoscale ice crystals in situ with electron microscopy methods is very challenging, arising from the electron beam radiation, low image contrast, potential impurities in the sample and others. It is clear that the authors in this work made a lot of effort, however, this reviewer have major concerns as listed in the following:

Response to the general comment:

We thank Reviewer #3 for the insightful comments and raising major points for improving the validity of our experiments. We have done our best to provide discussions on the reviewer's points based on experimental data and add additional supplementary information that may address their concerns.

Comment #1:

1). In-situ movies are not available for this review; they should be provided. Not sure whether it is the review portal issues, or missed submission.

Response #1:

We thank the reviewer for pointing this out. We have included the *in-situ* movies of growth of nanocrystalline ice as supporting movies that we have used for our quantification process.

Added movie (Supplementary Movie 1)

Original text (Main text, page 7, lines 160-162)

The growths of individual ice I_c and ice I_{c+h} nanocrystal domains were tracked with *in-situ* imaging by obtaining time-series images of ice particles growing on the amorphous ice film (Methods).

Revised text (Main text, page 8, lines 190-192)

The growths of individual ice I_c and ice I_{c+h} nanocrystal domains were tracked with *in-situ* imaging by obtaining time-series images of ice particles growing on the amorphous ice film (**Supplementary Movie 1 and Methods**).

Comment #2:

2). In Figure 1, the amorphous structure under continuous electron beam irradiation showed nucleation and growth of ice nanocrystals at 94k. The authors also showed under lower temperature 143k and after annealing, crystalline ice with different structures were found. The slow and fast growth are not clearly distinguished. And, the electron beam effects, and the difference from the annealing are not well explained.

Response #2:

We appreciate the reviewer for raising important points on differentiating slow and fast-growing ice nanoparticles and regarding electron beam damage. We clarify that our experimental results shown in Fig. 1 were performed by imaging multiple holes in a sequential manner, instead of irradiating a single area multiple times. This reduces the electron beam effects in our observation of the heating-induced crystallization.

The slow and fast growths of ice I_c and ice I_{c+h} domains, respectively, are distinguished by the apparent difference in the extents of growth of each type of domain, measured from the time-resolved TEM observations in Fig. 3e. In Fig. 3e, the slowly growing domains (shown in orange) are initially small, exhibit relatively little growth, and remain small in the final frame. The fast growing domains (shown in blue) are either relatively large initially, or the domain areas increase at steeper rates than those characterized as slow-growth. The phases of these domains exhibiting different growth rates have been assigned by electron diffraction (Fig 1i,j and Supplementary Fig. 10). For the slow-growing domains, which ultimately remain small even after long periods of annealing, TEM imaging and SAED as shown in Fig. 1i reveal the characteristic ice I_c peaks. Additionally, the size distribution of the ice I_c domains in the TEM image of in Fig. 1i is shown in the added Supplementary Fig. 9. For the fast-growing domains, the diffraction patterns exhibit crystalline peaks and streaks, which are features of hetero-nanocrystals with defects (Fig. 1j and Supplementary Fig. 10). Hence, the fact that slow-growing and fast-growing particles exhibited significantly different growth rates and sizes at the final frame of the time-resolved observation allowed us to distinguish the slow and fast growths characteristic of I_c and I_{c+h} domains, respectively.

To address the reviewer's comment on the effect of electron beam damage on amorphous ice, we have performed control experiments and found that heating-induced and beam-induced crystallization processes are manifested differently in our experimental conditions. We have also identified the dose limits for each of our imaging conditions to ensure that our TEM data is reliable. Heating-induced ice crystallization, shown in the added Supplementary Fig. 3a produces ice domains that are mainly higher in contrast than the amorphous ice, with the larger domains having white-contrast regions surrounding them. For these results, we have imaged different locations of ice films which were exposed to different annealing times at 143 K to minimize the electron beam exposure. For electron beam-induced crystallization at 93 K before raising the temperature of the ice (Supplementary Fig. 3b), dark and bright spots, marked with blue and yellow arrows respectively, start to form on the ice film at a threshold dose of $137 \text{ e}^- \text{ \AA}^{-2}$. These spots eventually grow into domains that exhibit either black or white contrast compared to the amorphous portion of the film. The contrast differs according to the alignment of the domains with respect to the electron beam, which depends on the diffraction condition. Previous studies on heating-induced crystallization of amorphous ice have reported that the

increased mobility of the molecules throughout the entire ice film beyond 136 K lowers the thermodynamic driving force for nucleation and growth, resulting in crystallization^{10,11}. Electron beam-induced crystallization, on the other hand, is caused by (1) enhanced mobility of water molecules due to added beam energy and (2) local heating effects⁸, which leads to amorphous ice crystallizing at faster rates in the affected regions.

To understand the combined effects of heating and beam-induced crystallization, and to evaluate the dose limit, we have performed electron beam irradiation experiments at 143 K as shown in Supplementary Fig. 3c. At a threshold dose of $22.2 \text{ e}^- \text{ \AA}^{-2}$, dark spots due to beam-induced crystallization start to form amidst amorphous ice, that are distinct from domains formed from heat-induced crystallization. Using these results on the dose limit, we have excluded TEM data obtained at 143 K that has been exposed to more than $22.2 \text{ e}^- \text{ \AA}^{-2}$ of electron dose in the main text to ensure reliability of the data.

We have added a description for distinguishing the slow-growing ice I_c and the fast-growing I_{c+h} domains in the Main text. We have also added a discussion on the differences between heating-induced ice crystallization at 143 K and beam-induced ice crystallization in the Main text with more detail in the Supplementary information.

Added Figure (Supplementary information, page 15, lines 191-194)

Supplementary Fig. 9: Size distribution of small, slowly growing ice domains

The size distribution histogram was obtained from the image in Fig. 1h, characterized as ice I_c .

Added Figure (Supplementary information, page 9, lines 143-150)

Supplementary Fig. 3: Comparison between heating-induced and beam-induced crystallization.

a Heating-induced crystallization at 143 K, in which each image was obtained at different locations of the film to minimize electron beam exposure. **b** Beam-induced crystallization at 93 K, caused by continuous irradiation of the electron beam on the amorphous ice film. **c** Beam-induced crystallization at 143 K, examining the combined effects of heating amorphous ice to 143 K and the electron beam. Scale bars = 100 nm.

Original text (Main text, page 8, lines 175-177)

Ice I_{c+h} domains were classified by the domains that eventually reach beyond 10^4 nm^2 , and ice I_c by the domains that grew to no larger than $7,000 \text{ nm}^2$ at the last frame of observation (Fig. 3e).

Revised text (Main text, page 9, lines 205-210)

~~Ice I_{c+h} domains were classified by the domains that eventually reach beyond 10^4 nm^2 , and ice I_c by the domains that grew to no larger than $7,000 \text{ nm}^2$ at the last frame of observation (Fig. 3e).~~ The slow and fast growths of ice I_c and ice I_{c+h} domains, respectively, are apparent by the difference in the extents of growth of each type of domain, measured from the time-resolved

TEM observations in Fig. 3e. The domains of ice I_c (shown in orange) are initially small, exhibiting relatively little growth until the final frame. By contrast, the domains of ice I_{c+h} (shown in blue) are larger and expand mostly at steeper rates than those characterized as I_c .

Added section and text (Supplementary information, page 3-4, line 37-67)

A subsection in the supplementary text titled 1.1 Electron beam effects on amorphous ice film crystallization was added.

Added text (Main text, page 4, lines 88-91)

With this process, we observed heating-induced crystallization of amorphous ice, free from beam-induced crystallization of amorphous ice films that usually occurs in the prolonged exposure of ice films to irradiation (Supplementary Figs. 3, 4 and Supplementary text 1.1).

Comment #3:

3). The ice crystals of different atomic structures are shown in Figure 2. The quality of the images is not good enough to directly compare with atomic positions/structures in the atomic models. It is well known that the different sample thickness can change the high resolution images significantly.

Response #3:

We appreciate the reviewer for bringing up this point about the HRTEM results. In response to the reviewer's comment that the quality of the image is insufficient to resolve the atomic positions, we replaced the TEM image in Figs. 2e and 2f with low-pass filtered images that would allow us to resolve the features (added Supplementary Fig. 12a). We have also included a new set of TEM simulations to take into account the effect of sample thickness on our images. We have performed extensive TEM image simulations on cubic, hexagonal, and stacking defect models where the thickness and defocus values of the sample had been varied to find conditions that match the TEM images in Supplementary Fig 12b. We specifically target Areas 1, 2 and 3 marked in Supplementary Fig. 12b, because these are the regions we had indexed to find cubic, hexagonal, and stacking defect sequences, respectively, in Fig. 2e and f of the Main text. Representative images of TEM simulation results for the sequences with stacking defects, modelled by using Area 3 marked in Supplementary Fig. 12b, are shown in Supplementary Fig. 12c. As a result, we found that the stacking faults of ice crystals shown in Area 3 of Supplementary Fig. 12b are well-manifested in the simulated TEM images with sample thickness of 100 nm, defocus values of 80 to 90 nm, and a tilt of 1.5 mrad. Using these simulation conditions, we have performed TEM simulations on Areas 1, 2 and 3, which contains the regions in which we marked the atomic positions for I_c , I_h , and I_{c+h} sequences, respectively. We found that the experimental data matches well with our simulated TEM data (Supplementary Figs. 12d and e), and with our imaging conditions, the atom positions are

bright in contrast as the white spots correspond to the atomic positions in the exit wave images (Supplementary Fig. 12f). Through these newly performed simulation results that better match our experimental conditions, we can conclude that the white spots in our high resolution TEM images can be compared to the atomic positions in our atomic models (Supplementary Fig. 12g), showing the presence of cubic, hexagonal sequences, as well as stacking defects in heterocrystalline ice.

Through HRTEM results, our aim was to verify the heterocrystallinity and the presence of defects in the ice nanocrystals. For this purpose, we have obtained additional high-magnification TEM images of fast-growing ice I_{c+h} nanocrystals as shown in the top panel of added Supplementary Fig. 13. The presence of heterocrystallinity and defects, manifested by the streaks in the FFT patterns at the bottom panel of Supplementary Fig. 13, is common for all three of the images shown. Therefore, we believe that our HRTEM results are sufficient to conclude that the fast-growing ice nanocrystals are heterocrystalline in nature, possessing stacking defects.

Original Figure (Main text, Figure 2, page 19, lines 419-428)

Fig. 2: The structure of the ice I_{c+h} domain. **a**, HRTEM image of a grown hetero-crystalline ice I_{c+h} domain at the $[110]$ zone axis. **b,c**, FFT of the boxed areas in **a** at a region of ice I_c sequences along $[110]$ zone axis (**b**), and a hetero-crystalline region (**c**). **d**, IFFT of the $(\bar{2}20)$ peak of the white boxed region in **a**, revealing positions of non-cubic sequences. **e,f**, Zoomed-in images of the region marked with the orange dotted line showing I_c sequences (**e**), and with the blue dotted line showing I_h sequences and stacking defects (**f**). **g-i**, Models constructed for ice I_c (**g**), ice I_h (**h**), and ice I_{c+h} (**i**) sequences. **j-l**, TEM simulations results of ice I_c (**j**), ice I_h (**k**), and ice I_{c+h} (**l**) models. **m-o**, Configurations of ice I_c (**m**), ice I_h (**n**), and ice I_{c+h} (**o**) crystal structures with labelled planes. Scale bars = 2 nm.

Revised Figure (Main text, Figure 2, page 18, lines 358-367)

Fig. 2: The structure of the ice I_{c+h} domain. **a**, HRTEM image of a grown hetero-crystalline ice I_{c+h} domain at the $[110]$ zone axis. **b,c**, FFT of the boxed areas in **a** at a region of ice I_c sequences along $[110]$ zone axis (**b**), and a hetero-crystalline region (**c**). **d**, IFFT of the $(\bar{2}20)$ peak of the white boxed region in **a**, revealing positions of non-cubic sequences. **e,f**, Zoomed-in, **low-pass filtered** images of the region marked with the orange dotted line showing I_c sequences (**e**), and with the blue dotted line showing I_h sequences and stacking defects (**f**). **g-i**, Models constructed for ice I_c (**g**), ice I_h (**h**), and ice I_{c+h} (**i**) sequences. **j-l**, TEM simulations results of ice I_c (**j**), ice I_h (**k**), and ice I_{c+h} (**l**) models. **m-o**, Configurations of ice I_c (**m**), ice I_h (**n**), and ice I_{c+h} (**o**) crystal structures with labelled planes. Scale bars = 2 nm.

results of ice I_c (j), ice I_h (k), and ice I_{c+h} (l) models. **m-o**, Configurations of ice I_c (m), ice I_h (n), and ice I_{c+h} (o) crystal structures with labelled planes. Scale bars = 2 nm.

Added Figure (Supplementary information, page 18, lines 203-220)

Supplementary Fig. 12: Experimental and simulated HRTEM images determining the structural features in heterocrystalline ice. **a** The raw, original high-resolution TEM image of a grown heterocrystalline ice I_{c+h} domain at the $[110]$ zone axis, with cubic (yellow circles), hexagonal (green circles), and stacking defect sequences (cubic and hexagonal) marked, as presented in the original manuscript. **b** Low-pass filtered high-resolution image to reduce high-frequency noise. Areas marked with the white dotted square have been compared with TEM simulations to verify the accuracy of the atomic position labels in **a**. **c** Representative TEM simulated images for different ice slab thickness and defocus values, modelled with the image shown in Area 3. Thickness of 100 nm and defocus values ranging from 80 to 90 nm show the best match to the experimental TEM image of Area 3. TEM simulations were performed with the multislice algorithm, with $C_s = 1.5$ mm and 1.5 mrad tilt. **d,e,f,g** Results of TEM simulations. Cubic (yellow circles), hexagonal (green circles), and stacking defect sequences (cubic and hexagonal) are labelled. **d** False-colored images of Areas 1, 2 and 3 labelled in **b**. **e**

TEM simulated images, performed with thickness of 100 nm and defocus ranging from 80 to 85 that best match the images in **d**. **f** Exit wave images indicating the positions of oxygen atoms in white. **g** Models used for simulation. Scale bars = 2 nm.

Added Figure (Supplementary information, page 19, lines 221-226)

Supplementary Fig. 13: High resolution TEM images of heterocrystalline ice.

Top) False-colored high resolution images of fast-growing heterocrystalline ice, showing lines of different contrast indicating the presence of heterocrystallinity and defects. Bottom) Corresponding FFT obtained from TEM images. The presence of streaks in the FFT indicate the presence of stacking defects.

Added text (Supplementary information, pages 5-6, lines 94-114)

A subsection in the supplementary text titled 1.3 TEM simulations for interpretation of HRTEM images was added.

Original text (Main text, pages 7, lines 148-151)

Models of ice I_c, ice I_h, and ice I_{c+h} were generated using molecular dynamics simulations (Figs. 2g-i). The models and the TEM simulations show similar images to the local structures visualized with the high-magnification TEM images (Figs. 2j-l and Methods).

Revised text (Main text, page 8, lines 172-179)

Models of ice I_c, ice I_h, and ice I_{c+h} were generated using molecular dynamics simulations (Figs. 2g-i). The models and the TEM simulations show similar images to the local structures visualized with the high-magnification TEM images (Figs. 2j-l and Methods). Additionally, extensive TEM simulations that match our experimental imaging conditions were performed for different thickness and defocus values of the proposed structures of ice at a region with a stacking fault, as discussed in more detail in Supplementary text 1.3 and Supplementary Fig. 12. The TEM simulation results verify that the atomic positions are manifested by light contrast in our imaging conditions.

Original text (Main text, page 7, lines 156-158)

This structure of ice consisting of both ice I_c and ice I_h sequences with stacking disorder have been proposed experimentally^{12,37} and by computational modeling^{23,38,44,45} in previous studies.

Revised text (Main text, page 8, lines 185-188)

This structure of ice consisting of both ice I_c and ice I_h sequences with stacking disorder was observed for multiple fast-growing ice domains in our HRTEM experiment as shown in Supplementary Fig. 13, which has been proposed experimentally^{12,37} and by computational modeling^{23,38,44,45} in previous studies.

Comment #4:

4). The image contrast of nanocrystals is very high. It is a concern that those nanoparticles can be from impurities. More analysis, such as EELS are needed to confirmed it is ice. In addition, chemical mapping is also necessary to exclude other possible origins.

Response #4:

We appreciate the reviewer for raising this point. First, we would like to point out the origin of the high contrast of the nanocrystals. We have utilized bright-field TEM (BFTEM) for imaging, which blocks the scattered electrons via an objective aperture and collects just the transmitted beam, which gives rise to higher diffraction contrast. We have provided a comparison between conventional TEM imaging and BFTEM to highlight the differences in particle contrast in the added Supplementary Fig. 6. Also, we note that a contrast-enhancement procedure was implemented for our data for better presentation shown in the added Supplementary Fig. 7, which have also allowed for binarization for the quantification of domain area.

Then, to confirm that the high-contrast features present at 143 K are indeed crystalline ice and not from impurities, we performed identical-location TEM. This process involves taking an image of a hole with amorphous ice film at 93 K, and then heating the sample to 143 K with the beam blanked, before taking another image of the same hole at 143 K. A representative image of the amorphous ice hole at 93 K exhibits uniform contrast, without features that may otherwise indicate the presence of impurities such as metal nanoparticles or ice deposits (added Supplementary Fig. 5a). The domains formed at 143 K are formed solely from amorphous ice. We have also observed some ice films containing very dark ice deposits at 93 K before the heating process. A representative example is shown in Supplementary Fig. 5b. We note that

images having these sorts of dark deposits were excluded from quantification. We have also utilized EDS elemental analysis to confirm that no other elements other than O from the ice, C from the carbon film, Au from the grid, Cu, Fe, and Co from the holder were present as potential sources of contaminants, such as salt contaminants (Supplementary Figs. 5c and d).

EELS spectroscopy also allows for the characterization of phases and mapping of elements. Supplementary Fig. 5e shows an ADF-STEM image for acquisition of low loss EELS spectra of amorphous ice with crystalline domains grown. Low loss EELS spectra were obtained in a region with an ice domain (green box and spectra). The domain region exhibits a shoulder peak at 9 eV in the core loss region, attributed to the electronic interband transition characteristic of ice. Core loss EELS mapping was also performed on an amorphous ice hole to rule out carbon-based contaminants. Three representative spectra obtained at different positions within a hole are shown in Supplementary Fig. 5f, in which we confirm that there are no C K edge peaks (284 eV) and only O K edge peaks (540 eV) from H₂O molecules are present.

Added figure (Supplementary information, page 12, lines 171-175)

Supplementary Fig. 6: Comparison between conventional TEM and BFTEM

Comparison between conventional TEM imaging and transmitted-beam imaging (bright field TEM, or BFTEM) shows that the latter exhibits higher contrast, aiding delineation of particles.

Added figure (Supplementary information, page 13, 177-180)

Supplementary Fig. 7: Contrast enhancement and binarization of ice nanoparticles.

Contrast enhancement and binarization process for analyzing ice nanoparticles. Scale bars = 200 nm.

Added Figure (Supplementary information, page 11, lines 159-170)

Supplementary Fig. 5 Elemental analysis of ice nanoparticles.

a,b Identical-location TEM results at 93 K (left) and 143 K (right). **a** shows a representative region used for quantitative analysis, and **b** shows a region with an adsorbed ice vapor contaminant, which was excluded during analysis. **c** HAADF-STEM image and **d** EDS spectrum obtained from elemental analysis of the region in **c**, showing that no other elements except for O from ice, C from the carbon film, Au from the grid, Cu, Fe, and Co from the holder. **e** ADF-STEM image of ice particles grown in amorphous ice (left) and low-loss spectra (right) obtained in the green box in **c** for a crystalline ice domain. The crystalline ice domain exhibits a shoulder peak at 9 eV, indicative of ice. **f** Core loss spectra at the region of the C K edge and O K edge, showing that no carbon peak is present and that there are no carbon contaminants. The inset shows the spectrum map.

Original text (Main text, page 4, lines 87-92)

As shown in the bright-field TEM images, individual ice nanoparticles appear as soon as the holder reaches 143 K which increase in number and change in area over time and consume the amorphous ice completely within 907 s (Fig. 1c). Growth kinetics of ice nanocrystals were investigated by measuring their areas and evaluating the number of domains and the crystallized fraction of amorphous ice over time (Methods)

Revised text (Main text, pages 4-5, lines 91-108)

As shown in the bright-field TEM (BFTEM) images of ice undergoing heating-induced crystallization, individual ice domains with dark contrast appear as soon as the holder reaches 143 K. The dark contrast originates from the transformation of amorphous ice into crystalline domains. This was revealed through identical-location TEM, in which a hole was imaged once at 93 K and again after the temperature was ramped to 143 K (Supplementary Fig. 5a). The images of amorphous ice before the formation of crystalline domains show uniform contrast without high-contrast features that may be indicative of any contaminants. The holes with adsorbed ice contaminants were excluded from quantification (Supplementary Fig. 5b). Additionally, through spectroscopic elemental analysis, we rule out the possibility of the dark contrast domains being contaminants (Supplementary Fig. 5c-f and Supplementary text 1.2). Accordingly, the dark contrast features originate from the crystallinity of the ice domains formed and they are clearly delineated in the bright-field TEM where the contrast of the ice domains is further enhanced (Supplementary Fig. 6).

The dark contrast ice domains increase in number and change in area over time and consume the amorphous ice completely within 907 s (Fig. 1c). Growth kinetics of ice nanocrystals were investigated by measuring their areas through a contrast enhancement procedure (Supplementary Fig. 7 and Methods) and evaluating the number of domains and the crystallized fraction of amorphous ice over time (Methods).

Added text (Supplementary information, pages 4-5, lines 70-92)

A subsection in the supplementary text titled 1.2 Elemental analysis of amorphous ice films and ice nanocrystals was added.

Comment #5:

5). The simulated conditions are not representative of the real conditions in this in situ Cryo-EM experiments. For example, at 94k the amorphous ice was found, which is not representative of the common scenario for ice structure formation.

Response #5:

According to the reviewer's comment, water at 143 K, which is the experimental condition, is in an ultraviscous liquid state and has movements that are too slow to undergo phase transitions. In Figure R1, we observed that the TIP4P/ICE model exhibits extremely subtle movements and a very slow ice growth rate around 153 K. To observe the growth characteristics of the water-

ice interface with limited computational resources up to a few microseconds, at least a temperature above 213 K are required. Therefore, we agree with the comment that our simulations are indicative of how ice may grow at the different ice surfaces. We have emphasized this point in the revised manuscript and provided further explanation regarding the temperature discrepancy with the experiment.

Figure R1, for reviewers only:

Figure R1: Simulation of the ice-water system from 133 K to 233 K.

a Diffusion coefficient of H₂O molecules in bulk water. **b** Ice growth rates on different ice surfaces: ice I_h (secondary prismatic) represented by the blue line, ice I_c {112} represented by the orange line, and the interface of ice I_h and ice I_c (basal) represented by the yellow line.

Original text (Main text, page 3, lines 69-72)

In this report, we use *in-situ* cryo-electron microscopy (cryo-EM) and molecular dynamics (MD) simulations to directly observe the time-resolved growth of individual ice nanocrystal polymorphs and investigate their respective interfacial structural dynamics.

Revised text (Main text, page 3, lines 67-70)

In this report, we use *in-situ* cryo-electron microscopy (cryo-EM) and molecular dynamics (MD) simulations for **modelling growths at different ice surfaces** to directly observe the time-resolved growth of individual ice nanocrystal polymorphs and investigate their respective interfacial structural dynamics.

Added text (Main text, page 10, lines 225-231)

The simulations were conducted at 233 K, under conditions where ice seeds with high curvature can grow, and water molecules possess adequate mobility for phase transitions. We note that higher temperatures increase the extent of water molecule diffusion and consequently the ice growth rate, and therefore, a higher growth rate of the ice crystal is expected compared to our experimental conditions (143 K). The simulations were modeled and interpreted as indications

of how different ice surfaces undergo crystallization.

References

1. Lim, K. *et al.* A Large-Scale Array of Ordered Graphene-Sandwiched Chambers for Quantitative Liquid-Phase Transmission Electron Microscopy. *Adv. Mater.* **32**, (2020).
2. Löfgren, P., Ahlström, P., Lausma, J., Kasemo, B. & Chakarov, D. Crystallization kinetics of thin amorphous water films on surfaces. *Langmuir* **19**, 265–274 (2003).
3. Smith, R. S., Huang, C., Wong, E. K. L. & Kay, B. D. Desorption and crystallization kinetics in nanoscale thin films of amorphous water ice. *Surf. Sci. Lett.* **367**, L13–L18 (1996).
4. Harada, K., Sugimoto, T., Kato, F., Watanabe, K. & Matsumoto, Y. Thickness dependent homogeneous crystallization of ultrathin amorphous solid water films. *Physical Chemistry Chemical Physics* **22**, 1963–1973 (2020).
5. Dohnálek, Z. *et al.* The effect of the underlying substrate on the crystallization kinetics of dense amorphous solid water films. *J. Chem. Phys.* **112**, 5932–5941 (2000).
6. Malek, J. The applicability of Johnson-Mehl-Avrami model in the thermal analysis of the crystallization kinetics of glasses. *Thermochim. Acta* **267**, 61–73 (1995).
7. Andrew H. N., and Molinero V. Identification of Clathrate Hydrates, Hexagonal Ice, Cubic Ice, and Liquid Water in Simulations: the CHILL+ Algorithm. *J. Phys. Chem. B* **119**, 9369-9376 (2015).
8. Xu, H., Ångström, J., Eklund, T. & Amann-Winkel, K. Electron Beam-Induced Transformation in High-Density Amorphous Ices. *J. Phys. Chem. B* **124**, 9283–9288 (2020).
9. Malkin, T. L., Murray, B. J., Brukhno, A. V, Anwar, J. & Salzmann, C. G. Structure of ice crystallized from supercooled water. *Proc. Natl. Acad. Sci. U.S.A.* **109**, 1041–1045 (2012).
10. Jenniskens, P. & Blake, D. F. Crystallization of Amorphous Water Ice in the Solar System. *ApJ* **473**, (1996).
11. Shephard, J. J. & Salzmann, C. G. Molecular Reorientation Dynamics Govern the Glass Transitions of the Amorphous Ices. *J. Phys. Chem. Lett.* **7**, 2281–2285 (2016).

Reviewer #1 (Remarks to the Author):

The authors have performed additional, detailed experiments and new data analysis. They have addressed points #1-#8 raised in my previous report. I think the manuscript can be published after addressing the following items -- I only have additional comments regarding the previous points #1 and #7.

Point #1: I was hoping that the films studied were $>10 \mu\text{m}$ -thick. Having thin films of $\sim 132 \text{ nm}$, as it is now reported in the manuscript, is relevant to interpreting the authors' experiments. I think the authors should make more evident that they are observing 3D-ice nucleation at approx $t < 215 \text{ sec}$ (with an Avrami exponent $n = 0.613$; ice particles $< 132 \text{ nm}$) and 2D-ice growth at $t > 215 \text{ sec}$ (with an Avrami exponent $n = 2.98$; large ice particles). The authors explain this in the supplementary information (SI) but the text remains unclear. I would recommend to include/rewrite lines 110-120 to include the text from their response letter (page 2) "While our amorphous ice film is estimated to be....axis will be halted."

- In line 121, the authors state that "...the first stage is characterized by the slow increase in the crystallized fraction, the second stage 121 by an acceleration...". Again, it should be stated that this first stage corresponds to 3D-ice nucleation while the remaining 2 stages correspond to 2D-ice growth.

- Can the authors clarify whether the 3D-ice particles ($< 132 \text{ nm}$) are always I_c ice (while ice particles of size approx $> 132 \text{ nm}$ are $I_c + I_h$ ice)?

- Related to this point, the title and/or abstract of the manuscript should stress that the authors are indeed studying the growth of 2D- and 3D-ices in thin films (and not just 3D ice nucleation). My point is that there are confining effects in this work that play an important role in the conclusions.

Point #7: I still have doubts regarding the conclusion that the density of liquid water next to the ice $I_c + I_h$ is depleted. Although the new Fig. 19 of the supplementary material shows a coarse-grained density that oscillates below 0.94 g/cc , this can be simply a lack of statistics. Note that the oscillations extend for only 1-2 Å. My suggestion is for the authors to be cautious and weaken this point.

Reviewer #2 (Remarks to the Author):

I am satisfied with the response and revision.

One minor comment: it will be nice to add the HRTEM of the interface to the figures in the main text.

Reviewer #3 (Remarks to the Author):

The revised manuscript addressed most of my previous concerns. I have no objection to publishing the revised manuscript in Nature Communications.

Response to Reviewer Remarks

Reviewer #1 (Remarks to the Author):

The authors have performed additional, detailed experiments and new data analysis. They have addressed points #1-#8 raised in my previous report. I think the manuscript can be published after addressing the following items -- I only have additional comments regarding the previous points #1 and #7.

Response to the general comment:

We appreciate the helpful comments that Reviewer #1 have provided with us, which improves the quality and reliability of our manuscript. We have done our best to address the additional points, as shown below.

Comment #1:

Point #1: I was hoping that the films studied were $>10 \mu\text{m}$ -thick. Having thin films of $\sim 132 \text{ nm}$, as it is now reported in the manuscript, is relevant to interpreting the authors' experiments. I think the authors should make more evident that they are observing 3D-ice nucleation at approx $t < 215 \text{ sec}$ (with an Avrami exponent $n = 0.613$; ice particles $< 132 \text{ nm}$) and 2D-ice growth at $t > 215 \text{ sec}$ (with an Avrami exponent $n = 2.98$; large ice particles). The authors explain this in the supplementary information (SI) but the text remains unclear. I would recommend to include/rewrite lines 110-120 to include the text from their response letter (page 2) "While our amorphous ice film is estimated to be....axis will be halted."

Response #1:

In response to the reviewer's comment on the significance of the relatively thin ice used for our experiment, we have added the above-mentioned sentence written in our previous response letter to our Main text with reference to Supplementary Fig. 8.

Added text (Main text, pages 5-6, lines 119-132)

While the amorphous ice film is estimated to be at a scale thick enough to consider the nucleation and crystallization as a bulk process, we note that once an ice domain's diameter exceeds around 132 nm , the thickness of the ice film, ice growth along the thickness axis of the ice film stops, but continues growth along the directions normal to the thickness (Supplementary Fig. 8a). This starts to occur after approximately 215 s of annealing, where ice nanocrystal diameters start to increase rapidly beyond around 300 nm (Supplementary Fig. 8b), becoming the dominant mechanism of crystallization. Initially, the fitted Avrami parameter is $n = 0.613$ indicating 3D-ice nucleation of the ice particles, and after 215 s the fitted Avrami parameter becomes $n = 2.98$, based on the log-log plot in Supplementary Fig. 8c, which indicates 2D-ice growth⁵⁰. Based on these results, we can conclude that, during crystallization,

nanocrystals first nucleate within amorphous ice, followed by their growth until around 132 nm, and if their diameters reach beyond 132 nm they would grow large enough to impinge on the surface and their growths along the thickness axis will be halted.

Comment #2:

- In line 121, the authors state that "...the first stage is characterized by the slow increase in the crystallized fraction, the second stage 121 by an acceleration...". Again, it should be stated that this first stage corresponds to 3D-ice nucleation while the remaining 2 stages correspond to 2D-ice growth.

Response #2:

We have added statements in the Main text regarding our interpretation of the general Avrami parameters, to make evident that we are observing 3D ice nucleation and subsequent 2D ice growth in our experimental conditions.

Added text (Main text, page 6, lines 136-137)

As mentioned previously, the first stage corresponds to three-dimensional ice nucleation, while the second and third stages correspond to two-dimensional ice growth.

Comment #3:

- Can the authors clarify whether the 3D-ice particles (<132 nm) are always I_c ice (while ice particles of size approx >132nm are I_c+I_h ice)?

Response #3:

We have obtained diffraction patterns for multiple individual ice particles and found that ice particles of areas less than around 7,000 nm², or around ~50 nm only show peaks that are present in ice I_c as shown in Supplementary Fig. 10a. Particles that exceed 132 nm in diameter have streaks in the diffraction patterns like those shown in Supplementary Fig. 10b. This indicates the presence of stacking defects and show that the domains are ice I_{c+h}.

Comment #4:

- Related to this point, the title and/or abstract of the manuscript should stress that the authors are indeed studying the growth of 2D- and 3D-ices in thin films (and not just 3D ice nucleation). My point is that there are confining effects in this work that play an important role in the conclusions.

Response #4:

We have revised the title, introduction, and the concluding paragraph of the manuscript to reflect our experimental conditions in thin amorphous ice films.

Original text (Title)

Observing growth and interfacial dynamics of nanocrystalline ice

Revised text (Title)

Observing growth and interfacial dynamics of nanocrystalline ice in thin amorphous ice films

Original text (Main text, page 3, lines 69-70)

... to directly observe the time-resolved growth of individual ice nanocrystal polymorphs and investigate their respective interfacial structural dynamics.

Revised text (Main text, pages 2-3, lines 72-74)

... to directly observe the time-resolved growth of individual ice nanocrystal polymorphs **on amorphous ice films with nanoscale thickness** and investigate their respective interfacial structural dynamics.

Original text (Main text, page 14, lines 330-332)

We used *in-situ* cryo-EM and MD simulations to track the early-stage growths of individual ice nanocrystal polymorphs to reveal their distinct growth dynamics and interfaces in the early stage of ice crystallization.

Revised text (Main text, page 15, lines 351-352)

In conclusion, we utilized cryo-EM and MD simulations to track the early-stage growths of individual ice nanocrystal polymorphs in an ice film of nanoscale thickness to reveal their distinct growth dynamics and interfaces in the early stage of ice crystallization.

Additionally, we have added in the abstract our observations of 3D and subsequent 2D ice growth in an amorphous ice film of nanoscale thickness.

Added text (Main text, page 2, lines 30-32)

we directly observe crystalline ice growth in an amorphous ice film of nanoscale thickness, which exhibits three-dimensional ice nucleation and subsequent two-dimensional growth.

Comment #5:

Point #7: I still have doubts regarding the conclusion that the density of liquid water next to the ice Ic+Ih is depleted. Although the new Fig. 19 of the supplementary material shows a

coarse-grained density that oscillates below 0.94 g/cc, this can be simply a lack of statistics. Note that the oscillations extend for only 1-2 Å. My suggestion is for the authors to be cautious and weaken this point.

Response #5:

We recognize that the structure of low-density water at the interface exists in a repeated and random distribution for a very short duration, and there may be simply a lack of statistics for the observed results. Following the reviewer's suggestion, we become more cautious about the low-density liquid water in the simulation results.

Original text (Main text, page 14, lines 336-339)

While water molecules near growing ice I_c crystals do not exhibit significant differences in the density or structure compared to bulk water, the fast-growing prismatic planes of ice I_{c+h} reveal the presence of a quasi-ice interface, a low-density region which possesses a degree of tetrahedral order.

Revised text (Main text, page 15, lines 360-361)

While water molecules near growing ice I_c crystals do not exhibit significant differences in the density or structure compared to bulk water, the fast-growing prismatic planes of ice I_{c+h} reveal the presence of a quasi-ice interface, a region that may correspond to LDL, with higher tetrahedral order and lower density than liquid water.

Reviewer #2 (Remarks to the Author):

I am satisfied with the response and revision.

One minor comment: it will be nice to add the HRTEM of the interface to the figures in the main text.

Response to Reviewer #2:

We appreciate Reviewer #2's comments that have significantly improved our manuscript.

We have included our observations of the interface using HRTEM in Fig. 4e,f. Using the HRTEM image shown in Fig. 4e, we have demonstrated that crystallinity extends up to the amorphous interfacial region, which decreases in contrast near the interfacial region which is amorphous in nature. The color-mapped inverse FFT image in Fig. 4f maps the areas that are crystalline, which allows us to also identify the areas which are amorphous by the pixel intensities of the inverse FFT.

Added text (Main text, page 12, lines 286-292)

A high-resolution TEM image that includes the surroundings of a heterocrystalline ice domain is shown in Fig. 4e. The atomic columns of white contrast indicate crystallinity from the ice domain, which exhibit I_c stacking sequences as well as planar defects. These atomic columns extend up to the amorphous interfacial region and decrease in contrast near the interfacial region. The inverse FFT image in Fig. 4f for the masked crystalline peaks (inset) reveal the region occupied by crystalline ice where pixel intensities are bright and color-mapped in yellow, and the interfacial region that lacks crystallinity which is closer to purple.

Original Figure (Main text, page 21, lines 384-394)

Fig. 4: The crystalline/amorphous ice interface. **a**, False color time-resolved *in-situ* TEM images of the first four frames of the growths of P1 and P2, which correspond to ice I_c and ice I_{c+h} nanocrystals, respectively. **b**, Intensity profiles of P1 and P2 in sequential frames of the TEM images. P2 shows the presence of a bright-contrast region indicative of low mass-thickness contrast. **c**, Contours of P1 and P2 and **d**, changes in their area over time. **e**, TEM image of a representative continuously-growing ice domain with its interfacial region. The white dashed circle represents the area where the aperture was inserted for SAED analysis. **f**, Diffraction patterns obtained at the aperture area and at amorphous ice at 143 K. **g**, Radial average of the amorphous components of the diffraction patterns in **f**. Scale bars for TEM images = 50 nm. Scale bars for diffraction patterns = 2 nm^{-1} .

Revised Figure (Main text, page 34, lines 736-750)

Fig. 4: The crystalline/amorphous ice interface. **a**, False color time-resolved *in-situ* TEM images of the first four frames of the growths of P1 and P2, which correspond to ice I_c and ice I_{c+h} nanocrystals, respectively. Scale bar = 50 nm. **b**, Intensity profiles of P1 and P2 in sequential frames of the TEM images. P2 shows the presence of a bright-contrast region indicative of low mass-thickness contrast. **c**, Contours of P1 and P2 and **d**, changes in their area over time. Scale bar = 50 nm. **e**, HRTEM image of a heterocrystalline ice domain that includes the surroundings of the domain. Stacking sequences are labelled. **f**, False-colored inverse FFT image, which shows areas that are high in crystallinity as yellow, and areas that are low in crystallinity in purple. Scale bars = 2 nm. **g**, TEM image of a representative continuously-growing ice domain with its interfacial region. The white dashed circle represents the area where the aperture was inserted for SAED analysis. Scale bar = 50 nm **h**, Diffraction patterns obtained at the aperture area and at amorphous ice at 143 K. **i**, Radial average of the amorphous components of the diffraction patterns in **h**, with the data points (solid dots) fitted with two Gaussian distributions (solid lines). Scale bar for diffraction patterns = 2 nm^{-1} .

Reviewer #3 (Remarks to the Author):

The revised manuscript addressed most of my previous concerns. I have no objection to publishing the revised manuscript in Nature Communications.

Response to Reviewer #3:

We appreciate Reviewer #3's comments that have significantly improved our manuscript.